# Darwinian Memory: A Training-Free Self-Regulating Memory System for GUI Agent Evolution

Hongze Mi [* 1]  Yibo Feng [* 1 2]  WenJie Lu [* 1]  Song Cao [* 3]  Jinyuan Li [3]  Yanming Li [1 4]  Xuelin Zhang [1]
Haotian Luo [4]  Songyang Peng [5]  He Cui [1]  Tengfei Tian [1]  Jun Fang [1]  Hua Chai [1]  Naiqiang Tan [1]

## Abstract

Multimodal Large Language Model (MLLM) agents facilitate Graphical User Interface (GUI) automation but struggle with long-horizon, cross-application tasks due to limited context windows. While memory systems provide a viable solution, existing paradigms struggle to adapt to dynamic GUI environments, suffering from a granularity mismatch between high-level intent and low-level execution, and context pollution where the static accumulation of outdated experiences drives agents into hallucination. To address these bottlenecks, we propose the Darwinian Memory System (DMS), a self-evolving architecture that constructs memory as a dynamic ecosystem governed by the law of "survival of the fittest." DMS decomposes complex trajectories into independent, reusable units for compositional flexibility, and implements Utility-driven Natural Selection to track survival value, actively pruning suboptimal paths and inhibiting high-risk plans. This evolutionary pressure compels the agent to derive superior strategies. Extensive experiments on real-world multi-app benchmarks validate that DMS boosts general-purpose MLLMs without training costs or architectural overhead, achieving average gains of 18.0% in success rate and 33.9% in execution stability, while reducing task latency, establishing it as an effective self-evolving memory system for GUI tasks.

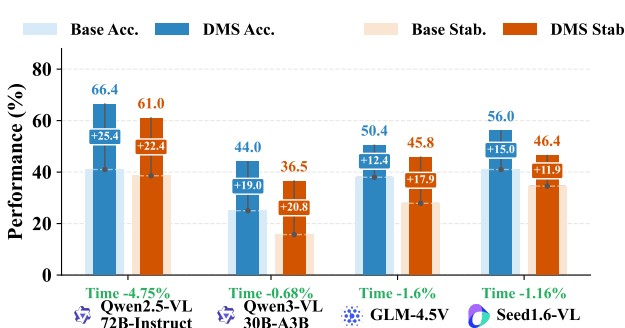

*Figure 1.* Performance Overview. In multi-app GUI scenarios, DMS consistently boosts Accuracy and Stability across diverse general-purpose models while reducing task latency.

## 1. Introduction

Recently, Multimodal Large Language Models (MLLMs) have fundamentally reshaped Graphical User Interface (GUI) automation by enabling visual perception (Wang et al., 2024a; Bai et al., 2025; Qin et al., 2025b; Cheng et al., 2024). However, constrained by context window limits, agents often suffer from context amnesia and hallucinations during long-horizon tasks. Memory systems (Wang et al., 2024b; Park et al., 2023; Wang et al., 2023; Zhao et al., 2024) have proven effective in bridging limited contexts via accumulated experiential knowledge. However, directly transposing these paradigms to the GUI domain encounters two critical bottlenecks:

**Rigidity of Monolithic Paradigms.** Existing systems typically operate on a holistic storage and retrieval basis, where long-horizon workflows are preserved as static, indivisible sequences (Dai et al., 2025; Li et al., 2025c). While high-level semantic matching effectively recalls relevant tasks, this approach tightly couples general intent with complete low-level execution paths. In complex GUI environments, this memory paradigm becomes brittle. Specifically, due to a lack of flexibility, excessive rigidity causes the entire fixed sequence to be invalidated by dynamic environmental changes.

**Stagnation in Static Storage.** Current agent systems operate on a static memory storage paradigm (Zhao et al., 2024;

---

[*]Equal contribution  [1]Didichuxing Co. Ltd  [2]The Chinese University of Hong Kong, Shenzhen  [3]Tianjin University  [4]Sun Yat-sen University  [5]Fudan University. Correspondence to: Naiqiang Tan <tannaiqiang@didiglobal.com>.

Salama et al., 2025), resulting in a repository cluttered with redundant data. Crucially, GUI environments are inherently non-stationary and evolving. Without a mechanism to verify temporal validity, suboptimal and outdated trajectories accumulate as toxic priors (Chhikara et al., 2025). Lacking evolutionary selection to distill experience, these systems fail to purge obsolete strategies or adapt to layout shifts, causing the agent to persist in inefficient behaviors. This stagnation hinders workflow optimization, leading to sustained inefficiency and high operational latency.

In nature, biological intelligence emerges not from static storage but from the continuous adaptation of populations. Species retain advantageous traits through natural selection while exploring new survival paths through mutation and eliminating unfit individuals. Inspired by this principle, we propose the **Darwinian Memory System (DMS)**, a self-evolving architecture integrated within a hierarchical Planner-Actor framework (Zhang et al., 2025a; Tan et al., 2024). We treat agent memory as a dynamic ecosystem governed by two core mechanisms.

Specifically, we fundamentally restructure the memory paradigm. Diverging from approaches that abstract entire long-horizon trajectories via summaries, we deconstruct workflows into combination of action subsequences. Functioning as independent, reusable units, this granularity allows historical experience to flexibly adapt to dynamic GUI sub-tasks. Furthermore, the reuse of verified units bypasses redundant reasoning for established patterns, significantly reducing inference costs and latency while enhancing execution stability. Throughout interaction, DMS enforces a survival-of-the-fittest dynamic by pruning long-tail entries based on a multi-factor survival value (i.e., popularity, decay, penalties). This effectively purges obsolete strategies and minimizes noise. Crucially, to prevent stagnation in local optima, DMS incorporates a probabilistic mutation mechanism. Instead of remaining static, even high-value memories are challenged, encouraging the agent to explore superior paths. Additionally, we employ dynamic Bayesian estimation to model reputation, transforming high-risk memories into evolutionary pressure. This compels the agent to discover optimal solutions, which are then captured as high-quality memories, closing the loop. Fundamentally, this mechanism empowers the agent to continuously evolve, achieving comprehensive gains in accuracy, stability, and efficiency as illustrated in Figure 1.

In summary, the main contributions of this paper are as follows:

- We introduce a structural memory paradigm that deconstructs monolithic workflows into reusable units. By decoupling intent from state, it resolves the rigidity of traditional paradigms and serves as the substrate for flexible composition and evolution.

- We propose Darwinian Memory System (DMS), a self-evolving ecosystem that autonomously prunes toxic priors to prevent context pollution and adapt to environmental shifts, generating evolutionary pressure that compels the agent to continuously optimize strategies.

- Extensive experiments demonstrate that DMS significantly boosts the performance of general-purpose models, achieving superior task success rates and execution efficiency.

## 2. Related Work

### 2.1. GUI Agents

MLLMs augment GUI agent automation by leveraging advanced multimodal perception (Achiam et al., 2023; Team et al., 2024; Anthropic, 2024; Bai et al., 2025). To enhance interaction capabilities, research has focused on large-scale supervised fine-tuning (SFT) and reinforcement learning (RL) using diverse datasets (Hong et al., 2024; Xu et al., 2025; Gu et al., 2025; Liu et al., 2025; Bai et al., 2024). Addressing long-horizon tasks, recent works employ hierarchical frameworks to decompose queries into sub-goals (Agashe et al., 2025; Zhang et al., 2025a; Kapoor et al., 2024; Tan et al., 2024; Wang et al., 2024a), or introduce specialized visual grounding modules for precise execution (Gou et al., 2024; Cheng et al., 2024). However, in long-horizon or cross-application scenarios, these methods lack sustainable mechanisms to manage extensive interaction histories, inevitably leading to context amnesia and reasoning fragmentation over time.

### 2.2. Memory and Experience Learning

Memory mechanisms transform agents into lifelong learners (Park et al., 2023; Packer et al., 2023). In GUI agents, Retrieval-Augmented Generation (RAG) is prevalent, utilizing historical trajectories as in-context demonstrations (Lai et al., 2025; Zheng et al., 2023). Recent research emphasizes active evolution (Huang et al., 2025a; Yu et al., 2025; He et al., 2025). EchoTrail-GUI (Li et al., 2025b) and MobileUse (Li et al., 2025a) leverage autonomous exploration to filter data. Advanced frameworks like R2D2 (Huang et al., 2025b) and H2R (Ye et al., 2025) use dynamic replay and hindsight reflection to refine decision-making, while MemEvolve (Zhang et al., 2025b) adapts the memory system via meta-evolution. Nevertheless, most paradigms rely on monolithic storage, creating rigid memories that fracture under dynamic layout shifts. Furthermore, lacking survival-based selection, toxic priors and obsolete entries accumulate unchecked.

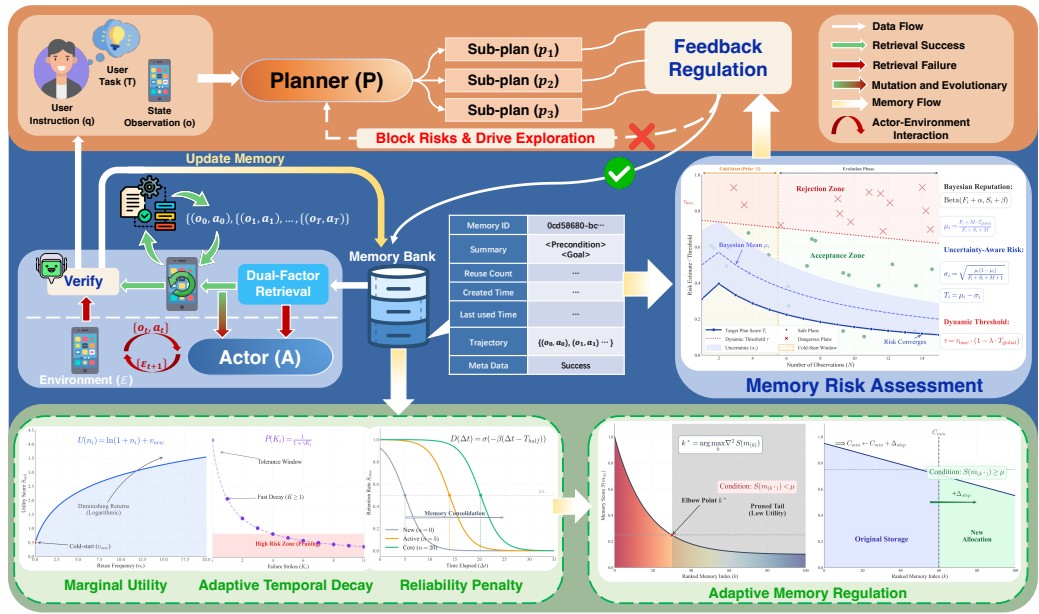

*Figure 2.* Overview of the Darwinian Memory System. The architecture integrates a Planner-Actor framework with a self-evolving memory mechanism. The Feedback Regulation module (Top) suppresses low-reputation plans via Bayesian risk assessment to drive exploration. The Memory Bank (Center) supports In-place Evolutionary Updates, where mutated trajectories with higher efficiency overwrite existing entries. To ensure system stability, Homeostatic Regulation (Bottom) dynamically prunes obsolete memories using the Elbow Method based on a multi-factor survival value (i.e., utility, decay, and reliability).

## 3. Method

### 3.1. Problem Formulation

The GUI terminal environment is modeled as a deterministic transition function $\mathcal{E}$. Let $s_t \in \mathcal{E}$ denote the state of the environment at a discrete time step $t$. The environment evolves according to the transition dynamics $s_{t+1} = \mathcal{E}(s_t, a_t)$, where $a_t \in \mathcal{A}$ represents an atomic action executed by the agent, and $\mathcal{A}$ denotes the feasible action space.

To rigorously evaluate DMS, we establish a canonical Planner-Actor framework without specialized GUI architectures, denoted as policy $\pi$. This framework consists of a high-level Planner ($P$) and a low-level Actor ($A$). Both modules are conditioned on a task-agnostic natural language system prompt $q$. The interaction process operates as a nested loop driven by the high-level task $T$:

**Planning Phase.** At the beginning of a planning cycle $t$, the agent receives an observation $o_t$ derived from the current state $s_t$. The Planner $P$ decomposes the high-level task $T$ (or the current progress) into a sequence of executable sub-tasks based on the $o_t$ and $q$:

$$P(T, o_t, q) = \{p_1, p_2, \ldots, p_k\}, \quad \text{where } k \leq 5$$

Here, $\{p_i\}_{i=1}^k$ represents a short-horizon trajectory of sub-goals intended to guide the Actor.

**Execution Phase.** The Actor $A$ processes the sub-tasks sequentially. For a specific sub-task $p_i$, the Actor generates an atomic action $a_t \in \mathcal{A}$ at each step based on the current

observation and the sub-task instruction $a_t = A(o_t, q, p_i)$.

Upon executing $a_t$, the environment transitions to $s_{t+1} = \mathcal{E}(s_t, a_t)$. This execution loop for $p_i$ continues until either the sub-task is successfully resolved (verified by the Planner or a heuristic) or a local step limit $Max_A$ is reached.

**Replanning and Termination.** The control flow returns to the Planner under two conditions: (1) the Actor successfully completes the entire sequence $\{p_k\}$, or (2) the Actor fails on any sub-task $p_i$. The Planner then regenerates a new plan based on the updated observation. This hierarchical cycle repeats until the global task $T$ is completed or the global step limit $Max_P$ is exceeded.

### 3.2. Mechanism of Darwinian Memory System

This section details the DMS, following the system's operational workflow: (1) Memory Construction (§3.2.1) outlines the methodology for memory formation; (2) Memory Utilization (§3.2.2) describes how memory functions within the agent; (3) Self-Regulation (§3.2.3) presents the core mechanism for long-term memory regulation; and (4) Risk Assessment (§3.2.4) establishes a feedback-driven evolutionary mechanism. Additionally, we analyze the key factors governing the system's steady-state equilibrium in Appendix C.

#### 3.2.1. MEMORY CONSTRUCTION

Classical memory paradigms construct historical experience by summarizing long atomic trajectories, which, in GUI

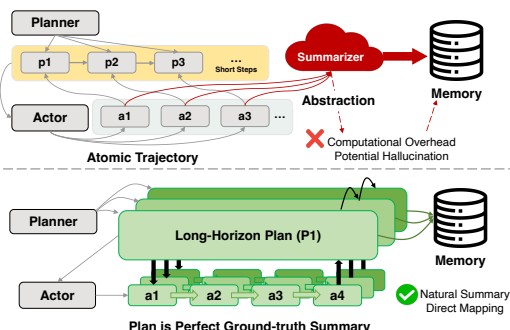

*Figure 3.* Memory construction overview. Unlike short-step methods requiring expensive post-hoc summarization (a), our approach (b) leverages long-horizon plans as natural, ground-truth summaries for atomic actions.

scenarios, incurs high latency, hallucination risks, and low value density (Figure 3). To address this, we reconstruct the memory paradigm by prompting the Planner to formulate coarse-grained sub-tasks $p_i$. These sub-tasks serve as intrinsic ground-truth summaries, eliminating hallucination risks and memory rigidity, while yielding independent and highly flexible memory units.

Formally, we enforce a structured format for each sub-task to maximize robustness: $p_i = \langle Precondition, Goal \rangle$. The *Precondition* describes the requisite UI state, while the *Goal* defines the target state transformation. Consequently, a memory entry $m$ is constructed as a tuple:

$$m = (p, \tau, s_{meta})$$

where $p$ is the natural language plan serving as the semantic index, $\tau = \{(o_0, a_0), ..., (o_T, a_T)\}$ is the dense execution trajectory performed by the Actor, and $s_{meta}$ includes metadata such as execution success status and descriptions. To prevent memory fragmentation, trajectories with $|\tau| = 1$ (i.e., single atomic actions) are filtered out. This ensures that the memory system exclusively retains non-trivial behavioral patterns that contribute to the long-term evolution.

### 3.2.2. MEMORY UTILIZATION MECHANISM

DMS employs Dual-Factor Retrieval to achieve precise memory matching, while leveraging $\epsilon$-Mutation and Evolutionary Replacement strategies to enable the continuous update and evolution of the memory system itself.

**Dual-Factor Retrieval.** Before the Actor commences a sub-task, the system queries the memory index using the current planner instruction $\hat{p} = \langle \hat{p}_{pre}, \hat{p}_{goal} \rangle$. To enhance robustness against state perturbations, we use Dual-Factor Similarity Metric that decouples the matching of preconditions and goals:

$$\text{Score}(\hat{p}, p) = \text{sim}(\phi(\hat{p}_{\text{pre}}), \phi(p_{\text{pre}})) \cdot \text{sim}(\phi(\hat{p}_{\text{goal}}), \phi(p_{\text{goal}}))$$

where $\phi(\cdot)$ denotes the embedding function and $sim(\cdot, \cdot)$ represents cosine similarity. This multiplicative formulation ensures that a memory is retrieved only when both

the starting state context and the intended objective align, significantly reducing false positives.

DMS adopts a decoupled storage architecture to accelerate retrieval: high-dimensional trajectories $\tau$ are persisted on disk, while their semantic summaries p are encoded as dense index vectors. Upon a retrieval hit, the Actor reuses the stored $\tau$. This mechanism yields a trifecta of benefits: drastically reducing computational latency and token costs, while ensuring deterministic execution stability by eliminating generation variance.

**$\epsilon$-Mutation and Evolutionary Replacement.** Relying solely on retrieved memories may trap the agent in suboptimal local optima. To ensure policy flexibility and continuous improvement, we introduce $\epsilon$-Mutation Strategy. Even when a high-confidence memory is retrieved, the agent acts exploratively with a small probability $\epsilon$:

$$\pi_{\text{exec}}(s_t) = \begin{cases} \text{Replay}(\tau_{\text{retrieved}}), & p = 1 - \epsilon \\ \text{Actor}(s_t, q)(\text{Mutation}), & p = \epsilon \end{cases}$$

During mutation, Actor re-attempts sub-task from scratch. If the new trajectory $\tau'$ proves successful and more efficient ($|\tau'| < |\tau|$), it triggers an in-place evolutionary update, overwriting the existing entry. This mechanism continuously drives the memory ecosystem towards optimal efficiency.

### 3.2.3. SELF-REGULATION STRATEGY

Unbounded memory accumulation inevitably leads to storage saturation and retrieval latency due to a diluted signal-to-noise ratio. To maintain system efficiency, we introduce self-regulation strategy that autonomously prunes suboptimal trajectories. We define the survival value $S(m_i)$ as a composite metric of utility, temporal decay, and reliability:

**Marginal Utility with Cold-Start Protection.** We model memory utility $U(n_i)$ as a function of reuse frequency $n_i$ that exhibits diminishing marginal returns, while incorporating a grace period to shield nascent memories from premature pruning:

$$U(n_i) = \ln(1 + n_i) + v_{\text{new}}$$

where $v_{new}$ is boost factor for recent memories, ensuring young candidates survive the initial selection pressure.

**Adaptive Temporal Decay.** We introduce a Sigmoid-based temporal decay component $D(\Delta t, n_i)$ to explicitly penalize dormancy. This term assigns significantly lower scores to memories that remain inactive for extended periods:

$$D(\Delta t, n_i) = \frac{1}{1 + e^{\beta(\Delta t - T_{\text{half}}(n_i))}}$$

Here, $\Delta t$ represents the logical time steps elapsed since the last retrieval. The dynamic half-life $T_{\text{half}}(n_i)$ extends logarithmically with usage:

$$T_{\text{half}}(n_i) = T_{\text{base}} + \mu \cdot \ln(1 + n_i)$$

where $T_{\text{base}}$ is the baseline retention span, and $\mu$ controls the sensitivity of memory consolidation. This mechanism ensures that the system naturally depreciates obsolete trajectories, prioritizing recent and frequently accessed knowledge.

**Reliability Penalty.** To enable self-correction, we incorporate a penalty term driven by a post-execution verification mechanism (details in Appendix D). Let $K_i$ denote the accumulated count of verification failures for memory $m_i$. The reliability factor $P(K_i)$ acts as a suppression coefficient:

$$P(K_i) = \frac{1}{1 + \gamma K_i}$$

where $\gamma$ is the penalty severity coefficient. This term rapidly degrades the score of error-prone trajectories, facilitating their removal during the pruning phase.

Combining these components, the final value $S(m_i)$ is computed as:

$$S = \underbrace{[\ln(1 + n_i) + V_{\text{new}}]}_{\text{Utility}} \cdot \underbrace{\left[\frac{1}{1 + e^{\beta(\Delta t - T_{\text{half}}(n_i))}}\right]}_{\text{Adaptive Decay}} \cdot \underbrace{\left[\frac{1}{1 + \gamma K_i}\right]}_{\text{Reliability}}$$

This multidimensional metric ensures memory system prioritizes memories that are frequently reused, recently active, and highly reliable, forming the basis for our subsequent pruning algorithm.

**Adaptive Memory Regulation.** Based on the ranked utility curve $f(k) = S(m_{(k)})$, we employ a distribution-aware mechanism to regulate memory capacity. We utilize the Elbow Method to identify the optimal cutoff index $k^*$ by maximizing the discrete second-order gradient, which highlights the transition to the long tail:

$$k^* = \arg\max_k \nabla^2 f(k)$$

Pruning is triggered upon reaching the preset capacity $C_{min}$. However, we incorporate a safeguard against high-quality saturation. If the distribution exhibits low curvature and the elbow score $f(k^*)$ exceeds the population mean, the system identifies the overflow as valuable experience:

$$\text{if } f(k^*) \geq \mu \implies C_{\text{min}} \leftarrow \min(C_{\text{min}} + \Delta_{\text{step}}, C_{\text{max}})$$

In this scenario, we trigger expansion, extending the storage limit instead of pruning.

### 3.2.4. RISK ASSESSMENT AND FEEDBACK REGULATION

To prevent the agent from repeatedly exploring known failure modes and to accelerate evolutionary convergence, we introduce the negative feedback regulation mechanism. It assesses the risk of generated plans via Bayesian inference.

**Bayesian Reputation Modeling.** The failure tendency of plan $p_i$ is $\theta_i \in [0, 1]$. Given observations $D_i$ with $F_i$ failures and $S_i$ successes, we employ a Beta-Binomial conjugate prior model. The posterior distribution is updated as:

$$p(\theta_i|D_i) = \text{Beta}(\theta_i|F_i + \alpha, S_i + \beta)$$

where $\alpha, \beta$ are global smoothing priors. By defining the prior strength $M = \alpha + \beta$ and the global failure rate $T_{\text{global}} = \alpha/(\alpha + \beta)$, the expected failure probability $\mu_i$ is derived using Bayesian smoothing:

$$\mu_i = \frac{F_i + M \cdot T_{\text{global}}}{F_i + S_i + M}$$

**Uncertainty-Aware Risk Scoring.** To mitigate the "Cold-Start" problem, we incorporate the posterior standard deviation $\sigma_i$ as an uncertainty metric:

$$\sigma_i = \sqrt{\frac{\mu_i(1 - \mu_i)}{F_i + S_i + M + 1}}$$

We define the final Risk Score $T_i$ using a Lower Confidence Bound approach: $T_i = \mu_i - \sigma_i$.

**Dynamic Thresholding.** To balance exploration with safety, the rejection threshold $\tau$ adapts to ecosystem health, tightening constraints under high global error rates:

$$\tau = \tau_{\text{base}} \cdot (1 - \lambda \cdot T_{\text{global}})$$

where $\lambda$ denotes penalty sensitivity. Plans exceeding $\tau (T_i > \tau)$ are suppressed, effectively pruning maladaptive mutations. This unsupervised filtration incentivizes the exploration of viable strategies, fostering the autonomous refinement of agent logic.

## 4. Experiments

### 4.1. Experiments Setting

We evaluate the performance of DMS on dynamic task execution using the widely adopted AndroidWorld benchmark. The specific implementation details are as follows.

**AndroidWorld.** For dynamic task execution, we evaluate DMS on AndroidWorld (Rawles et al., 2024), an online benchmark that runs in live Android emulator. It contains 116 distinct tasks across 20 real-world applications. Through parameter randomization, these tasks yield millions of unique variants, rigorously testing a model's adaptability to diverse user instructions and dynamic UI states.

**Settings & Baselines.** We establish baselines using high-performance general-purpose MLLMs across varying scales. Specifically, we employ Qwen2.5-VL-72B-Instruct (Bai et al., 2025), Qwen3-VL-30B-A3B-Instruct (Team, 2025), and GLM-4.5V (Team et al., 2025) as open-source backbones, alongside Seed1.6-VL (Guo et al., 2025) as the closed-source representative. Open-source models are deployed on a server with 8× NVIDIA A100 80G GPUs. To validate effectiveness, we benchmark DMS against a comprehensive suite of recent GUI methods. Detailed prompt templates are provided in Appendix H. Full experimental setups are detailed in Appendix B.

*Table 1.* Success Rate (%) on the AndroidWorld benchmark. We compare methods across four attributes: Unified Backbone (simplicity), Open Weights (accessibility), Training-Free (efficiency), and Evolutionary Memory (mechanism novelty). PA-Lite denotes our canonical Planner-Actor baseline, while DMS-PA-Lite is the baseline augmented with DMS.

| Method | Univ. | Weights | Train. | Mem. | Model | SR↑ |
|---|---|---|---|---|---|---|
| GPT-4o (Achiam et al., 2023) | ✓ | ✗ | ✓ | ✗ | GPT-4o | 34.5 |
| Aguvis (Xu et al., 2025) | ✓ | ✗ | ✗ | ✗ | GPT-4o + Aguvis | 37.1 |
| UGround (Gou et al., 2025) | ✗ | ✗ | ✓ | ✗ | GPT-4o | 44.0 |
| Aria-UI (Yang et al., 2025) | ✓ | ✗ | ✗ | ✗ | GPT-4o + Aria-UI | 44.8 |
| AndroidGen (Lai et al., 2025) | ✗ | ✗ | ✓ | ✓ | GPT-4o | 46.8 |
| EchoTrail-GUI (Li et al., 2025c) | ✗ | ✗ | ✓ | ✓ | GPT-4o | 51.7 |
| Gemini (Team et al., 2024) | ✓ | ✗ | ✓ | ✗ | Gemini-1.5-Pro | 22.8 |
| Claude (Anthropic, 2024) | ✓ | ✗ | ✓ | ✗ | Claude Computer-Use | 27.9 |
| Agent-S2 (Agashe et al., 2025) | ✓ | ✗ | ✓ | ✓ | Claude-3.7-Sonnet | 54.3 |
| V-Droid (Dai et al., 2025) | ✗ | ✓ | ✗ | ✓ | V-Droid | 59.5 |
| Seed1.5-VL (Guo et al., 2025) | ✓ | ✓ | ✓ | ✗ | Seed1.5-VL | 62.1 |
| Aguvis (Xu et al., 2025) | ✓ | ✓ | ✗ | ✗ | Qwen2-VL-72B-Instruct[†] | 26.1 |
| Qwen2.5-VL (Bai et al., 2025) | ✓ | ✓ | ✓ | ✗ | Qwen2.5-VL-72B-Instruct | 35.0 |
| EchoTrail-GUI (Li et al., 2025c) | ✗ | ✓ | ✓ | ✓ | Qwen2.5-VL-72B-Instruct | 46.6 |
| UI-TARS (Qin et al., 2025a) | ✓ | ✓ | ✗ | ✗ | Qwen2.5-VL-72B-Instruct[†] | 46.6 |
| MobileUse (Li et al., 2025a) | ✗ | ✓ | ✓ | ✓ | Qwen2.5-VL-72B-Instruct | 62.9 |
| UI-Venus (Gu et al., 2025) | ✓ | ✓ | ✗ | ✗ | Qwen2.5-VL-72B-Instruct[†] | 65.9 |
| PA-Lite | ✓ | ✓ | ✓ | ✗ | Qwen3-VL-30B-A3B-Instruct | 25.0 |
| PA-Lite | ✓ | ✓ | ✓ | ✗ | GLM-4.5V | 38.0 |
| PA-Lite | ✓ | ✗ | ✓ | ✗ | Seed1.6-VL | 41.0 |
| PA-Lite | ✓ | ✓ | ✓ | ✗ | Qwen2.5-VL-72B-Instruct | 41.0 |
| **DMS-PA-Lite** | ✓ | ✓ | ✓ | ✓ | Qwen3-VL-30B-A3B-Instruct | 44.0 (↑19.0) |
| **DMS-PA-Lite** | ✓ | ✓ | ✓ | ✓ | GLM-4.5V | 50.4 (↑12.4) |
| **DMS-PA-Lite** | ✓ | ✗ | ✓ | ✓ | Seed1.6-VL | 56.0 (↑15.0) |
| **DMS-PA-Lite** | ✓ | ✓ | ✓ | ✓ | Qwen2.5-VL-72B-Instruct | **66.4** (↑25.4) |

## 4.2. Main results

Table 1 demonstrates that DMS-PA-Lite establishes a new state-of-the-art on the AndroidWorld benchmark, validating the quantitative superiority of our approach. Our method, powered by Qwen2.5-VL-72B, achieves a success rate of 66.4%, significantly outperforming both proprietary frontier models like GPT-4o (34.5%) and specialized agents like UI-Venus (65.9%). Furthermore, this performance advantage is universal across diverse architectures, yielding consistent gains such as +15.0% for Seed1.6-VL and +25.4% for Qwen2.5-VL-72B. This data confirms that the Darwinian Memory System effectively unlocks the latent planning capabilities of MLLMs, transforming open-source models into competitive agents regardless of their native scale.

Despite operating as a strictly zero-shot, training-free framework, DMS-PA-Lite surpasses fully fine-tuned models like UI-TARS (46.6%). This confirms that DMS bridges the performance gap between open-weight models and their closed-source counterparts, enabling efficient deployment without the need for domain-specific pre-training.

## 4.3. Memory Reuse over Long Horizons

To investigate the temporal evolution of the Darwinian Memory System, we visualize memory reuse rates across five execution rounds in Figure 4. A distinct ascending gradient is evident across all models, where reuse rates climb from a minimal cold-start in R1 to over 30% in later stages (e.g., Qwen2.5-VL-72B rises from ∼12% to >30%). This

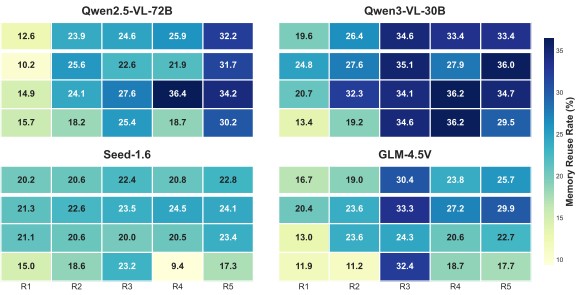

*Figure 4.* Heatmap of Memory Reuse Rates. The visualization displays the memory reuse probability across 5 execution rounds for three models. Darker colors indicate higher reuse rates.

trend empirically validates our evolutionary mechanism: as the agent explores, high-quality trajectories are naturally selected and stored, effectively transforming historical experience into accessible computational assets. Notably, reuse rates stabilize between 30% and 36%, a structural ceiling imposed by our Dual-Factor Retrieval to prevent script overfitting. This design choice ensures a strategic balance, preserving the planner's flexibility for dynamic contexts. Furthermore, this behavior remains consistent across diverse architectures, from Qwen3-VL-30B to GLM-4.5V.

## 4.4. Success Rate Detail across rounds and difficulties

To quantify the impact of experience accumulation, we conducted a 5-round experiment comparing PA-Lite against DMS-PA-Lite. As illustrated in Figure 5a, DMS induces a distinct upward trajectory in success rates as tasks accu-

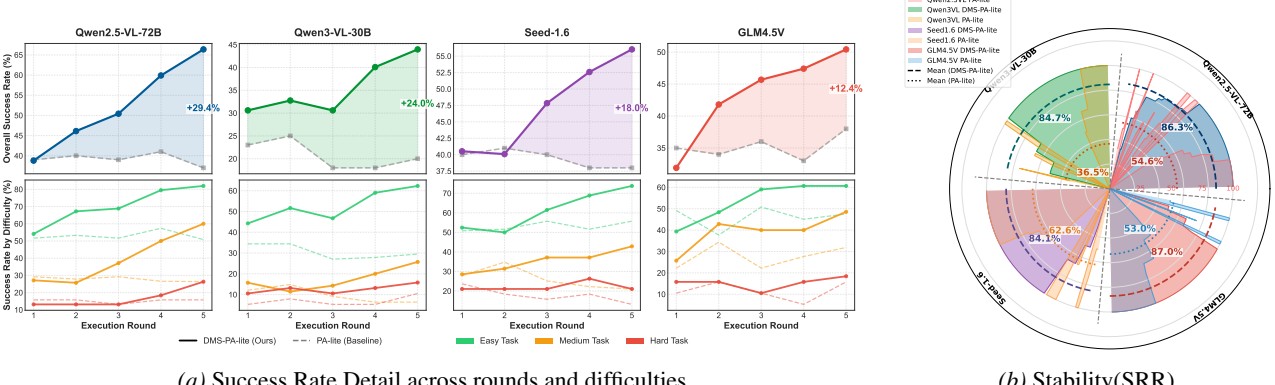

*(a)* Success Rate Detail across rounds and difficulties    *(b)* Stability(SRR)

*Figure 5.* Performance and Stability Analysis. (a) Success rate comparison of various open-source models with and without DMS across multiple experimental trials. The lower panel provides a granular breakdown of performance across varying task difficulties. (b) Stability landscape visualizing the Success Rate Retention (SRR) of different methods, distinguished by color. Please refer to Appendix E.2 for the formal definition of SRR.

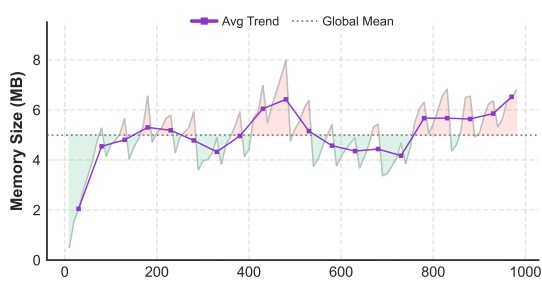

*Figure 6.* Visualization of Memory Maintenance Mechanism.

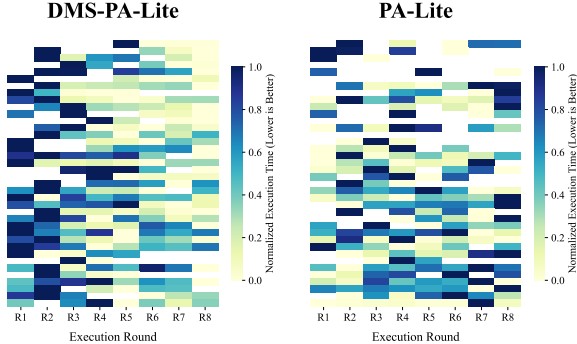

*Figure 7.* Heatmap visualization of normalized execution time across sequential rounds. Lighter hues represent reduced latency, while white cells denote failed trials, which are omitted from the time efficiency comparison.

mulate, contrasting sharply with the stagnation observed in the baseline. Specifically, Qwen3-VL-30B and Qwen2.5-VL-72B achieve cumulative gains of +24.0% and +29.4%, respectively, with consistent improvements across varying task difficulties. Notably, minor fluctuations observed in Qwen3-VL (Round 3) and Seed-1.6 (Round 2) are attributed to the feedback regulation mechanism; by suppressing high-risk plans, the system temporarily induces volatility during the agent's re-exploration phase.

Beyond raw success rates, DMS fundamentally transforms agent robustness, evaluated via the Stability Reference Rate (SRR). Baselines without DMS exhibit severe volatility across rounds, indicating that in the absence of accumulated experience, a single success does not guarantee reproducibility. In contrast, by effectively reusing verified memories, DMS drastically reduces the inherent probabilistic variance of MLLMs. This stability is evidenced by substantial SRR improvements, with Qwen3-VL-30B rising from 36.5% to 84.7% and GLM-4.5V from 53.0% to 87.0% (Figure 5b).

### 4.5. Dynamics of Memory Accumulation

To evaluate the long-term storage efficiency of the Darwinian Memory System, we tracked the memory footprint over 1,000 planning cycles (Figure 6). The results show a pe-

riodic oscillation around a mean of 5 MB, consistently peaking below 8 MB. Unlike systems that exhibit linear growth, our approach actively regulates capacity by identifying and purging obsolete trajectories during scheduled maintenance intervals. This autonomous balance between retention and pruning prevents the accumulation of redundant data, ensuring sustainable operation on resource-constrained devices.

### 4.6. Temporal Efficiency Analysis

Figure 7 visualizes the normalized task execution time across 8 sequential rounds, revealing a distinct contrast between the two paradigms. The DMS-PA-Lite panel (Left) exhibits a clear transition from high-latency states (dark blue) to optimized execution (light yellow) as experience accumulates. This trend empirically confirms that the retrieval of "Precondition-Goal" macros effectively bypasses the heavy computational load of MLLM reasoning, resulting in a consistent reduction in temporal overhead. Conversely, the PA-Lite baseline (Right) displays a stochastic distribution characterized by persistent dark blocks and erratic

fluctuations throughout the session. Lacking a mechanism to cache and reuse procedural knowledge, the baseline relies on repetitive chain-of-thought generation, leading to sustained inefficiency and high variance in operational latency.

## 5. Ablation

### 5.1. Ablation Study

To rigorously evaluate the contribution of individual DMS components, we conducted ablation studies using Qwen2.5-VL-72B as the backbone. We compared our full DMS-PA-Lite against five variants: (1) w/o Feedback Regulation: removing the success-rate tracking mechanism; (2) w/o Dynamic Thresholding: employing a fixed rejection threshold; (3) Standard Goal-Based Key: replacing our dual-factor key ($p_i = \langle p_{\text{pre}}, p_{\text{goal}} \rangle$) with the traditional goal-only format ($p_i = \langle p_{\text{goal}} \rangle$); (4) w/o Self-Regulation: disabling the evolutionary pruning of trajectories; and (5) PA-Lite: the standalone agent without DMS.

*Table 2.* Ablation study of DMS components on AndroidWorld. "Overall" denotes the success rate across all tasks.

| Method | Easy | Medium | Hard | Overall |
|---|---|---|---|---|
| **DMS-PA-Lite (Ours)** | **82.0%** | **60.0%** | **26.3%** | **66.4%** |
| w/o Feedback Regulation | 56.6% | 39.4% | 13.2% | 40.1% |
| w/o Dynamic Thresholding | 51.6% | 37.2% | 7.9% | 36.6% |
| Standard Goal-Based Key | 45.9% | 24.5% | 15.8% | 29.7% |
| w/o Self-Regulation | 52.5% | 37.2% | 26.3% | 39.2% |
| PA-Lite (Baseline) | 57.4% | 26.6% | 15.8% | 41.0% |

As presented in Table 2, every component is critical to the system's efficacy. We highlight follow key findings:

**Role of Feedback Regulation.** Removing Feedback Regulation leads to a 26.3% performance drop. This aligns with expectations, as without this mechanism, the system exerts no evolutionary pressure on the agent, causing performance to revert to the level of the PA-Lite baseline.

**Necessity of Dynamic Thresholding.** Removing dynamic thresholding causes a sharp performance drop of 29.8%. Without adaptive tolerance, the agent becomes brittle: valid plans are permanently inhibited due to early failures, while accumulated errors fail to reach the static suppression threshold, rendering the feedback loop ineffective.

**Risk of Conventional Paradigms.** Notably, the Standard Goal-Based Key method yields the lowest performance (29.7%), significantly underperforming even the memory-free baseline (41.0%). This confirms that applying rigid, goal-only retrieval in dynamic GUIs induces negative transfer, where context pollution from mismatched trajectories severely impairs reasoning.

**Efficiency of Self-Regulation.** Disabling self-regulation degrades accuracy by 27.2% due to the accumulation of

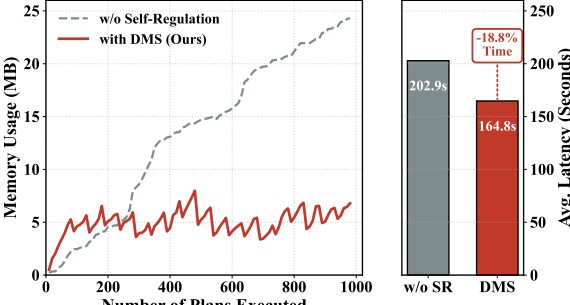

*Figure 8.* Impact of Self-Regulation on System Efficiency. Left: Cumulative memory usage (MB) plotted against the number of executed plans. Right: Bar chart displaying the average inference latency (seconds) per task for both configurations.

obsolete memories. Furthermore, as illustrated in Figure 8, we monitored memory storage footprint after completing 1,000 tasks. In the absence of Self-Regulation, memory accumulates unchecked, resulting in a 258% increase in disk usage compared to DMS. Interestingly, we observe that the memory growth rate without Self-Regulation is actually lower than that of DMS; this phenomenon is attributed to the accumulation of obsolete memories, which obstructs the update mechanism and decelerates expansion. Moreover, the excessive accumulation of inefficient memory inflates retrieval latency and induces hallucinations, thereby prolonging agent trajectories. This leads to an 18.8% increase in average execution time compared to DMS. These empirical results validate Self-Regulation as a cornerstone of DMS, essential for balancing resource efficiency with robust execution capabilities.

## 6. Conclusion

In this paper, we introduce the DMS, a memory system that fundamentally constructs agent memory as a dynamic, evolutionary ecosystem. By deconstructing monolithic workflows into combinatorial action subsequences and enforcing a survival-of-the-fittest mechanism, DMS enables continuous self-optimization in complex GUI environments. The integration of probabilistic mutation and Bayesian risk assessment effectively resolves the tension between stability and plasticity, allowing agents to escape local optima while maintaining robust execution. Empirical results demonstrate that DMS confers comprehensive advantages to general-purpose MLLMs, yielding significant improvements in success rates, stability (SRR), and inference efficiency. This work establishes a viable lifelong learning paradigm, paving the way for biologically inspired architectures.

## Impact Statement

This work aims to advance the capabilities of large language model-based digital agents. While such agents can significantly assist users with tasks on both mobile and desktop environments, they may not guarantee perfect execution accuracy. Furthermore, ensuring robust safety and user privacy throughout their operation remains an ongoing challenge in the field.

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

# Appendix

## A. Action Space

To ensure precise and effective task execution, we define a constrained action space. This approach simplifies the decision-making process by enabling the agent to ground its reasoning in a well-structured set of operations. The complete action space, detailing the parameters and description for each action, is summarized in Table 3. Each action type has certain parameters and detailed in description.

*Table 3.* Agent Action Space, Descriptions, and Arguments.

| Agent Action | Action Details | |
| --- | --- | --- |
|  | **Arguments** | **Description** |
| swipe | $start_x, start_y, end_x, end_y, duration_{ms}$ | Swipe from the given start coordinates to the given end coordinates. |
| input_text | $text, clear$ | Input the given text into a focused input field. If clear true, clears the input field before typing. |
| press_key | $press\_key$ | Enter the given keycode. |
| tap | $index, x, y, duration_{ms}$ | Tap or Long Press a UI element or coordinate. |
| start_app | $package$ | Start the given app. |
| remember | $information$ | Remember the given information. This is used to store information in the tool's memory. |
| complete | $success, reason$ | Complete the tool. This is used to indicate that the tool has completed its task. |

## B. Experiment Setting & Baseline

**Experiment Setting.**   All experiments are conducted using a multi-round protocol to simulate the long-term execution of repetitive daily tasks in realistic GUI environments. In our main result, we adopt distinct reporting metrics to ensure a rigorous comparison. For the memory-free baseline (PA-Lite), which lacks experience accumulation, we report the peak performance achieved across all rounds to represent its theoretical upper bound. Conversely, for DMS-PA-Lite, we report the final round performance to accurately reflect its true, evolved proficiency after experience accumulation. In addition, we provide an Extended Performance Analysis in Appendix G and a detailed Case Study in Appendix F.

**Hyperparameters Setting.** To ensure reproducibility, we detail the specific parameter configurations used in our DMS implementation. For the Survival Value ($S$) calculation, we adopted a balanced configuration to maintain ecosystem stability, setting the novelty bonus $V_{\text{new}} = 1.0$, the base protection period $T_{\text{base}} = 30.0$, the longevity coefficient $\alpha = 15.0$, the decay steepness $\beta = 0.5$, and the penalty coefficient $\gamma = 1.0$. Regarding the Dynamic Thresholding mechanism, we utilized a moderate sensitivity setting with $\lambda = 0.3$. It is worth noting that while these parameters serve as a robust baseline, achieving optimal performance may require fine-tuning based on the specific capabilities and error patterns of the underlying MLLM.

**Baselines.** On the AndroidWorld benchmark, we compare DMS against various state-of-the-art baselines from different model categories: (1) Training Method: Aguvis (Xu et al., 2025), Aria-UI (Yang et al., 2025), V-Droid (Dai et al., 2025), UI-tars (Gu et al., 2025), UI-Venus (Gu et al., 2025) (2) Framework Mehtod: GPT-4o (Achiam et al., 2023), UGround (Gou et al., 2025), AndroidGen (Lai et al., 2025), EchoTrail-GUI (Li et al., 2025c), Gemini (Team et al., 2024), Claude (Anthropic, 2024), Agent-S2 (Agashe et al., 2025), Seed1.5-VL (Guo et al., 2025), Qwen2.5-VL (Bai et al., 2025), MobileUse (Li et al., 2025a)

## C. Theoretical Analysis: Equilibrium and Purity

To rigorously evaluate the stability of the Darwinian Memory System, we model the memory dynamics as a stochastic flow process.

Let $N$ denote the total memory capacity, and $b_t = B_t/N$ be the proportion of bad trajectories at time step $t$. The system purity is defined as $Q_t = 1 - b_t$. We define the following parameters: $\alpha$ as the retrieval hit rate, $\epsilon$ as the mutation probability, and $p_{\text{fail}}$ as the inherent probability of generating a flawed trajectory during inference.

Contaminated memories are introduced into the system via two pathways: (1) inference misses where the agent generates a new but flawed plan, and (2) mutation steps where exploration fails. The total influx of contamination $\Phi_{\text{in}}^B$ is:

$$\Phi_{\text{in}}^B = \underbrace{(1 - \alpha) \cdot p_{\text{fail}}}_{\text{Miss \& Fail}} + \underbrace{\alpha \cdot \epsilon \cdot p_{\text{fail}}}_{\text{Mutation \& Fail}} = p_{\text{fail}} \cdot (1 - \alpha + \alpha\epsilon)$$

Conversely, contaminated memories are purged when the agent reuses a memory ($\alpha(1 - \epsilon)$), triggers the verification mechanism, and successfully identifies the error with probability $P_{\text{TP}}$. The outflow $\Phi_{\text{out}}^B$ is:

$$\Phi_{\text{out}}^B = \alpha(1 - \epsilon) \cdot b_t \cdot P_{\text{TP}}$$

Similarly, the flux dynamics for good memories, denoted as $\Phi_{\text{in}}^G$ and $\Phi_{\text{out}}^G$, are derived as:

$$\Phi_{\text{in}}^G = (1 - p_{\text{fail}}) \cdot (1 - \alpha + \alpha\epsilon),$$
$$\Phi_{\text{out}}^G = \alpha(1 - \epsilon) \cdot (1 - b_t) \cdot P_{\text{FN}}$$

where $P_{\text{FN}}$ represents the False Negative rate.

The system achieves homeostasis when the influx and outflow of memory types reach equilibrium ($\Phi_{\text{in}} = \Phi_{\text{out}}$). By solving the ratio, we derive the steady-state ratio of bad to good memories:

$$\frac{b_{\text{ss}}}{g_{\text{ss}}} = \left( \frac{p_{\text{fail}}}{1 - p_{\text{fail}}} \right) \cdot \left( \frac{P_{\text{FN}}}{P_{\text{TP}}} \right)$$

Consequently, the asymptotic purity $Q_{\text{ss}}$ of the memory system is:

$$Q_{\text{ss}} = \frac{1}{1 + \frac{b_{\text{ss}}}{g_{\text{ss}}}} = \frac{1}{1 + \mathcal{R}_{\text{fail}} \cdot \mathcal{R}_{\text{ver}}}$$

where $\mathcal{R}_{\text{fail}} = p_{\text{fail}}/(1 - p_{\text{fail}})$ is intrinsic failure odds of the LLM, and $\mathcal{R}_{\text{ver}} = P_{\text{FN}}/P_{\text{TP}}$ represents verification error ratio.

This derivation reveals a critical insight: maximizing system purity hinges on minimizing the verification error ratio $\mathcal{R}_{\text{ver}}$. Since $P_{\text{TP}}$ is typically high, the bottleneck lies in reducing $P_{\text{FN}}$. This provides the theoretical justification for the K-Verification Policy. By requiring $K$ accumulated strikes before deletion, the effective false negative rate decays exponentially:

$$P_{\text{FN}}^{\text{effective}} \approx (P_{\text{FN}})^K$$

Thus, even with a moderately accurate verifier, the system ensures $Q_{ss} \to 1$ as $K$ increases.

Finally, we analyze the rate of change $\Delta B_t = \Phi^B_{in} - \Phi^B_{out}$. By grouping terms, we observe:
$$\Delta B_t = C(b_t) + \epsilon \cdot [\alpha p_{fail} + \alpha b_t P_{TP}]$$
While $\epsilon$ cancels out in the steady-state ratio, it confirms that $\epsilon$ acts as a catalyst for evolutionary velocity. A higher $\epsilon$ accelerates the system's convergence to the steady state, validating our strategy of dynamic exploration to escape local optima rapidly.

The self-regulation strategy guarantees a compact, high-signal-to-noise ratio memory store, effectively balancing resource constraints with the continuous accumulation of procedural knowledge.

## D. Verification Design and Operational Mechanism

This section details the architectural design and operational logic of the verification mechanism within DMS.

### D.1. Theoretical Basis and Parameter Selection

As derived in our Theoretical Analysis (Appendix C), the reliability of the memory system is intrinsically linked to the rigor of the verification process. This relationship is quantified by the effective False Negative rate $P^{effective}_{FN} \approx (P_{FN})^K$. Here, $K$ represents the verification depth (or the number of independent verification checks). While a higher $K$ value theoretically enforces stricter validation and yields higher memory purity (information density), it inevitably incurs greater computational latency. To strike an optimal balance between memory quality and inference efficiency, we set $K = 3$ in our experimental implementation.

### D.2. The Verification Operational Mechanism

To operationalize this, we designed a comprehensive verification protocol that leverages the inherent multimodal reasoning capabilities of MLLMs. We adopt a strategy whereby the verifier prioritizes the logical consistency of the action history, rejecting the result when visual evidence explicitly contradicts the claimed success.

The specific system prompt used for the Verifier Agent is presented in Figure 9. Finally, the complete operational logic integrating this verification process into the DMS loop is formalized in Algorithm 1.

---

**Verifier Prompt**

**Role:** You are an expert Android Task Verifier. Your job is to determine if the agent's execution history successfully achieved the user's goal.
**Input Information:**

1. **Original Goal:** The user's original objective.

2. **Execution History:** The (Thought, Code) steps the agent *claims* it just performed. This is your **PRIMARY** source of truth.

3. **Final Screenshot:** The *ground truth* screenshot. This is your **SECONDARY** check for contradictions.

---

**YOUR VERIFICATION LOGIC (History-First):**

1. **Analyze History (Trust):** Read the Execution History. Did the agent perform the logical actions required to complete the Original Goal? (e.g., for "Save recording," did the agent tap('Save')?)

2. **Assume Success:** If the history looks correct, your default verdict is {"verified_success": true}.

3. **Visual Veto (Contradiction Check):** Now, look at the Final Screenshot. Does this screenshot *explicitly contradict* the agent's claim of success?

    - **Contradiction ($\to$ Fail):** The screenshot shows an *error message* (e.g., "Password incorrect").
    - **Contradiction ($\to$ Fail):** The screenshot shows the agent is in the *wrong application*.
    - **Contradiction ($\to$ Fail):** The goal was "Dismiss the 'OK' dialog," but the screenshot clearly shows the 'OK' dialog *is still visible*.
    - **NO Contradiction ($\to$ Success):** The goal was "Dismiss the 'OK' dialog," and the screenshot shows the dialog is *gone*. This **confirms** the history.
    - **NO Contradiction ($\to$ Success):** The goal was "Click the 'Save' button," and the screenshot shows the app has *moved to a different screen*. This **confirms** the history.

**Key Rule:** You must default to True (success) if the history is sound AND the screenshot does not provide *strong, undeniable proof* of failure.
**Output Format:** Respond ONLY with the JSON object: {"verified_success": <bool>, "reason": "<string>" }

*Figure 9.* The prompt for the Verifier Agent.

---

**Algorithm 1** DMS Verification Loop

---

1: **Input:** Global Task $T$, Memory Bank $\mathcal{M}$
2: **Param:** Max Steps $T_{\max}$, Smoothing Priors $\alpha$, $\beta$, Risk Threshold $\tau_{\text{risk}}$
3: **Init:** $\mathcal{L}_{\text{active}} \leftarrow \emptyset$;    $t \leftarrow 0$;    $R_{\text{task}} \leftarrow$ FAIL
4: **while** $T$ not completed **and** $t < T_{\max}$ **do**
5:      $s_t \leftarrow$ GetObservation()
6:      $\mathcal{P} \leftarrow$ Planner($s_t, T$)             # Generate sub-plans $\{p_1, ..., p_k\}$
7:      PlanFailed $\leftarrow$ FALSE
8:      **for all** $p_i \in \mathcal{P}$ **do**
9:          $m \leftarrow$ Retrieve($\mathcal{M}, p_i$)
10:          **if** $m \neq$ None $\wedge \rho_m < \tau_{\text{risk}} \wedge$ Random() $> \epsilon$ **then**    # Check Risk: $\rho_m$ derived from $\mu_m$ must be safe
11:             $\tau \leftarrow$ GetTrajectory($m$);    DoReuse $\leftarrow$ TRUE
12:          **else**
13:             $\tau \leftarrow$ ActorGenerate($s_t, p_i$);    DoReuse $\leftarrow$ FALSE
14:          **end if**
15:          $R_{\text{sub}} \leftarrow$ Execute($\tau$)             # Unified execution step
16:          **if** $R_{\text{sub}} ==$ SUCCESS **then**             # Case A: Success Handling
17:             **if** DoReuse **then**
18:                 $S_m \leftarrow S_m + 1$;    $\mathcal{L}_{\text{active}} \leftarrow \mathcal{L}_{\text{active}} \cup \{m\}$      # Reuse reward
19:             **else**
20:                 $m_{\text{new}} \leftarrow$ CreateMemory($p_i, \tau, S\leftarrow1, F\leftarrow0, K\leftarrow0$)
21:                 $\mathcal{M} \leftarrow \mathcal{M} \cup \{m_{\text{new}}\}$;    $\mathcal{L}_{\text{active}} \leftarrow \mathcal{L}_{\text{active}} \cup \{m_{\text{new}}\}$
22:             **end if**
23:          **else**                    # Case B: Failure Handling
24:             **if** DoReuse **then**
25:                 $K_m \leftarrow K_m + 1$             # Increment strikes
26:                 **if** $K_m \geq K_{\text{limit}}$ **then**
27:                    $\mathcal{M} \leftarrow \mathcal{M} \setminus \{m\}$          # Prune obsolete memory
28:                 **else**
29:                    $\mathcal{L}_{\text{active}} \leftarrow \mathcal{L}_{\text{active}} \cup \{m\}$      # Keep active for global penalty
30:                 **end if**
31:             **end if**
32:             PlanFailed $\leftarrow$ TRUE;    **Break**          # Discard & Replan
33:          **end if**
34:          Update $s_t, t$
35:      **end for**
36:      **if not** PlanFailed **and** CheckGlobalSuccess($s_t, T$) **then**
37:          $R_{\text{task}} \leftarrow$ SUCCESS;    **Break**
38:      **end if**
39: **end while**
40: **for all** $m \in \mathcal{L}_{\text{active}}$ **do**             # Global Feedback Regulation Stage
41:      **if** $R_{\text{task}} ==$ FAIL **then**             # Penalize only if global task failed
42:          $F_m \leftarrow F_m + 1$
43:      **end if**
44:      $\mu_m \leftarrow (F_m + \alpha)/(F_m + S_m + \alpha + \beta)$      # Update Mean $\mu_m$ and Risk Score $\rho_m$
45:      $\rho_m \leftarrow \mu_m - \sqrt{\mu_m(1 - \mu_m)/(F_m + S_m + \alpha + \beta + 1)}$
46: **end for**

---

# E. Evaluation Metrics

### E.1. Success Rate (SR)

Success Rate (SR) serves as the fundamental metric for evaluating the agent's overall efficacy in completing tasks within a specific experimental round. It is defined as the ratio of successfully completed tasks to the total number of tasks attempted.

Formally, let $\mathcal{T} = \{T_1, T_2, \dots, T_M\}$ denote the set of $M$ tasks in an experiment. Let $S_i \in \{0, 1\}$ represent the binary outcome of task $T_i$, where $1$ indicates success and $0$ indicates failure. The Success Rate is calculated as:

$$\text{SR} = \frac{1}{M}\sum_{i=1}^{M} S_i$$

A higher SR reflects the agent's general capability to satisfy user instructions across a diverse set of GUI scenarios.

### E.2. Success Retention Rate (SRR)

To rigorously evaluate the stochastic stability of the agent's correct reasoning, we introduce the Success Retention Rate (SRR). While standard accuracy measures the average performance, SRR specifically quantifies the agent's robustness against performance regression under repeated trials.

Formally, given a sequence of execution outcomes $x = \{x_1, \dots, x_N\}$, SRR is defined as the conditional probability that a

correct output ($x_t = 1$) is followed by another correct output ($x_{t+1} = 1$):

$$\text{SRR} = P(x_{t+1} = 1 \mid x_t = 1) = \frac{\sum_{t=1}^{N-1} \mathbb{I}(x_t = 1 \wedge x_{t+1} = 1)}{\sum_{t=1}^{N-1} \mathbb{I}(x_t = 1)}$$

where $\text{SRR} \in [0, 1]$. A higher SRR indicates a stable equilibrium, where the agent, once successful, is highly likely to reproduce the correct solution. A lower SRR indicates transient success, where correct outcomes are likely due to stochastic luck rather than robust capability.

### E.3. Memory Reuse Rate

To quantify the extent to which the agent leverages accumulated experience versus generating new actions from scratch, we introduce the Memory Reuse Rate. This metric measures the proportion of the action trajectory that is derived directly from the DMS memory retrieval.

For a completed task $T_i$, let the execution trajectory be a sequence of atomic actions $A_i = \{a_1, a_2, \ldots, a_{L_i}\}$, where $L_i$ is the total trajectory length. Let $A_i^{\text{mem}} \subseteq A_i$ denote the subset of actions that were instantiated via memory retrieval (i.e., executed as part of a retrieved macro-action or subsequence). The Memory Reuse Rate for the task set is defined as:

$$\text{MRR} = \frac{\sum_{i=1}^{M} |A_i^{\text{mem}}|}{\sum_{i=1}^{M} |A_i|}$$

where $|\cdot|$ denotes the count of atomic actions.

*Table 4.* Case Study of Longitudinal Stability. We report the detailed performance metrics across 5 consecutive rounds for representative cases characterized by performance fluctuations. **S**: Success (0/1), **T**: Time (seconds), **R**: Memory Reuse Rate (0-1).

| Task | Model | Diff | Round 1 | | | Round 2 | | | Round 3 | | | Round 4 | | | Round 5 | | |
|---|---|---|---|---|---|---|---|---|---|---|---|---|---|---|---|---|---|
| | | | S | T | R | S | T | R | S | T | R | S | T | R | S | T | R |
| BrowserDraw | Qwen2.5-72B | Easy | 0 | 226.9 | 0.00 | 0 | 279.9 | 0.58 | 1 | 112.3 | 0.07 | 1 | 496.0 | 0.36 | 1 | 154.7 | 0.67 |
| ExpenseAddMultiple | Qwen2.5-72B | Med | 1 | 339.2 | 0.10 | 0 | 779.4 | 0.07 | 0 | 342.2 | 0.05 | 1 | 328.1 | 0.11 | 1 | 295.6 | 0.13 |
| FilesDeleteFile | Qwen2.5-72B | Med | 1 | 335.9 | 0.00 | 1 | 317.2 | 0.11 | 1 | 151.7 | 0.40 | 0 | 220.0 | 0.00 | 1 | 301.9 | 0.50 |
| MarkorCreateNote | Qwen2.5-72B | Med | 1 | 197.0 | 0.09 | 1 | 437.3 | 0.13 | 0 | 111.6 | 0.41 | 1 | 256.7 | 0.64 | 1 | 103.2 | 0.67 |
| NotesRecipeCount | Qwen2.5-72B | Easy | 0 | 341.2 | 0.18 | 1 | 290.8 | 0.13 | 0 | 225.4 | 0.24 | 1 | 260.9 | 0.08 | 1 | 71.0 | 0.78 |
| OsmAndFavorite | Qwen2.5-72B | Med | 0 | 369.4 | 0.04 | 0 | 287.1 | 0.21 | 1 | 110.2 | 0.15 | 1 | 170.5 | 0.26 | 1 | 82.3 | 0.30 |
| RecipeDeleteDup | Qwen2.5-72B | Easy | 0 | 292.8 | 0.29 | 0 | 281.4 | 0.24 | 1 | 1349.5 | 0.37 | 0 | 166.3 | 0.28 | 0 | 175.1 | 0.26 |
| ClockStopWatch | Qwen3-30B | Easy | 0 | 43.5 | 0.00 | 1 | 20.8 | 0.00 | 0 | 68.8 | 0.00 | 1 | 21.6 | 0.00 | 1 | 18.1 | 0.08 |
| ContactsAddContact | Qwen3-30B | Easy | 1 | 63.6 | 0.00 | 1 | 75.5 | 0.00 | 0 | 65.1 | 0.50 | 1 | 65.2 | 0.00 | 1 | 72.9 | 0.50 |
| FilesDeleteFile | Qwen3-30B | Med | 1 | 166.6 | 0.00 | 1 | 126.3 | 0.29 | 0 | 204.6 | 0.48 | 0 | 141.7 | 0.60 | 1 | 115.5 | 0.75 |
| NotesIsTodo | Qwen3-30B | Easy | 1 | 55.2 | 0.00 | 1 | 47.9 | 0.50 | 0 | 159.7 | 0.67 | 1 | 48.6 | 0.61 | 1 | 46.1 | 0.50 |
| RecipeDeleteNoise | Qwen3-30B | Med | 1 | 510.3 | 0.00 | 1 | 423.4 | 0.09 | 0 | 562.8 | 0.05 | 0 | 377.8 | 0.39 | 1 | 238.9 | 0.06 |
| TasksDueOnDate | Qwen3-30B | Easy | 1 | 102.7 | 0.00 | 0 | 38.5 | 0.80 | 0 | 34.4 | 1.00 | 1 | 37.4 | 0.80 | 1 | 99.6 | 0.77 |
| NotesIsTodo | Seed1.6-VL | Easy | 1 | 114.9 | 0.00 | 0 | 146.6 | 0.33 | 1 | 144.0 | 0.00 | 1 | 132.1 | 0.00 | 1 | 155.3 | 0.00 |
| NotesTodoCount | Seed1.6-VL | Med | 1 | 168.5 | 0.54 | 0 | 91.2 | 0.43 | 1 | 103.2 | 0.00 | 1 | 119.9 | 0.00 | 0 | 96.3 | 0.67 |
| SimpleCalendar | Seed1.6-VL | Med | 0 | 68.5 | 0.00 | 1 | 131.4 | 0.00 | 0 | 325.6 | 0.00 | 1 | 117.8 | 0.00 | 1 | 137.7 | 0.00 |
| SystemWifiTurnOff | Seed1.6-VL | Easy | 1 | 47.1 | 0.57 | 0 | 35.0 | 0.67 | 1 | 50.5 | 0.40 | 1 | 89.9 | 0.29 | 1 | 51.2 | 0.50 |
| ContactsAddContact | GLM-4.5V | Easy | 1 | 186.6 | 0.00 | 0 | 138.3 | 0.00 | 1 | 139.6 | 0.00 | 1 | 141.1 | 0.00 | 1 | 123.4 | 0.00 |
| ExpenseAddSingle | GLM-4.5V | Easy | 0 | 598.9 | 0.08 | 1 | 499.5 | 0.00 | 1 | 710.9 | 0.17 | 0 | 554.4 | 0.05 | 1 | 519.3 | 0.00 |
| CalendarAddEvent | GLM-4.5V | Hard | 0 | 1010.3 | 0.00 | 1 | 392.4 | 0.00 | 0 | 764.4 | 0.90 | 1 | 523.3 | 0.05 | 1 | 471.1 | 0.21 |

## F. Case Study: Dynamics of Stability and Self-Regulation

Table 4 details specific task instances that exhibited performance volatility across the 5-round lifespan. While DMS significantly improves overall stability (as shown in SRR metrics), analyzing these fluctuations reveals the internal dynamics of the memory evolution process. We categorize these instability patterns into three distinct types:

**Exploration-Driven Fluctuation.** A common pattern involves an agent succeeding in early rounds, failing in intermediate rounds, and converging to a high-efficiency success in the final round. For instance, in *FilesDeleteFile* (Qwen2.5-72B), the agent succeeded in Rounds 1-3 but failed in Round 4, before achieving a stable success in Round 5 with 50% memory reuse. This reflects the feedback regulation mechanism at work: the system may proactively reject a suboptimal memory (causing a temporary failure or forced re-exploration) to escape a local optimum, ultimately discovering a more robust path.

Similarly, in *MarkorCreateNote*, the failure in Round 3 ($R = 0.41$) triggered a mutation that led to highly efficient successes in Rounds 4 and 5 ($R \approx 0.66$, Time reduced by $\sim 50\%$).

**Mitigation of Negative Transfer.** High memory reuse does not always guarantee success if the retrieved context is mismatched—a phenomenon known as negative transfer. A striking example is *TasksDueOnDate* (Qwen3-30B). In Round 3, the agent achieved a 100% reuse rate ($R = 1.0$) but failed the task ($S = 0$). This suggests the agent relied on a perfectly matching but factually incorrect memory (e.g., hallucinating a date based on history). However, DMS's survival mechanism successfully identified this failure. By Round 5, the agent corrected its behavior ($S = 1$), reducing reuse to a valid 77% ($R = 0.77$). This demonstrates DMS's ability to "unlearn" toxic high-confidence memories.

**Granularity Constraints and Convergence Latency.** Stability challenges persist in tasks necessitating high-granularity atomic actions or deep logical reasoning. For tasks like *ClockStopWatchPausedVerify* (Qwen3-30B) and *SimpleCalendarEventsOnDate* (Seed1.6-VL), the DMS proactively filters out single-step atomic memories to preserve the ecosystem's high information density. Consequently, in rapidly changing environments where execution relies on high-atomicity sequences (combinations of discrete steps), the Memory Reuse Rate remains at zero. This reflects a trade-off where DMS's efficacy is bounded when macro-action abstraction is inhibited by strict granularity requirements.

Furthermore, the temporal horizon required for convergence varies with task difficulty. As observed in *RecipeDeleteDuplicateRecipes* (Qwen2.5-72B), the agent struggles to maintain success despite moderate memory reuse. This suggests that for tasks demanding complex logical verification, a more prolonged phase of experience accumulation and path exploration is requisite before the system can effectively leverage memory for stable performance gains.

# G. Extended Performance Analysis

In the main paper (Table 1), we reported the peak performance (Best-of-$N$) for the memory-free baseline (PA-Lite) to compare DMS against the theoretical upper bound of the baseline's reasoning capabilities. However, given the stochastic nature of LLMs without memory, the average performance often reflects a more realistic expectation of daily utility. To provide a comprehensive view, Table 5 presents the average success rates for PA-Lite across multiple experimental rounds. As shown, when compared against the average baseline performance, DMS demonstrates even more substantial gains (e.g., ↑27.4% on Qwen2.5-VL compared to ↑25.4% against the peak). This indicates that DMS not only elevates the agent's capability ceiling but also effectively mitigates the performance variance inherent in memory-free architectures.

*Table 5.* Extended Analysis. Unlike the main text which reports the peak performance, this table presents the average success rate with standard deviation ($Mean \pm SD$) across 5 experimental rounds.

| Method | Univ. | Weights | Train. | Mem. | Model | Avg SR$_{\pm\sigma}$ |
|---|---|---|---|---|---|---|
| GPT-4o (Achiam et al., 2023) | ✓ | ✗ | ✓ | ✗ | GPT-4o | 34.5 |
| Aguvis (Xu et al., 2025) | ✓ | ✗ | ✗ | ✗ | GPT-4o + Aguvis | 37.1 |
| UGround (Gou et al., 2025) | ✗ | ✗ | ✓ | ✗ | GPT-4o | 44.0 |
| Aria-UI (Yang et al., 2025) | ✓ | ✗ | ✗ | ✗ | GPT-4o + Aria-UI | 44.8 |
| AndroidGen (Lai et al., 2025) | ✗ | ✗ | ✓ | ✓ | GPT-4o | 46.8 |
| EchoTrail-GUI (Li et al., 2025c) | ✗ | ✗ | ✓ | ✓ | GPT-4o | 51.7 |
| Gemini (Team et al., 2024) | ✓ | ✗ | ✓ | ✗ | Gemini-1.5-Pro | 22.8 |
| Claude (Anthropic, 2024) | ✓ | ✗ | ✓ | ✗ | Claude Computer-Use | 27.9 |
| Agent-S2 (Agashe et al., 2025) | ✓ | ✗ | ✓ | ✓ | Claude-3.7-Sonnet | 54.3 |
| V-Droid (Dai et al., 2025) | ✗ | ✓ | ✗ | ✓ | V-Droid | 59.5 |
| Seed1.5-VL (Guo et al., 2025) | ✓ | ✓ | ✓ | ✗ | Seed1.5-VL | 62.1 |
| Aguvis (Xu et al., 2025) | ✓ | ✓ | ✗ | ✗ | Qwen2-VL-72B-Instruct[†] | 26.1 |
| Qwen2.5-VL (Bai et al., 2025) | ✓ | ✓ | ✓ | ✗ | Qwen2.5-VL-72B-Instruct | 35.0 |
| EchoTrail-GUI (Li et al., 2025c) | ✗ | ✓ | ✓ | ✓ | Qwen2.5-VL-72B-Instruct | 46.6 |
| UI-TARS (Qin et al., 2025a) | ✓ | ✓ | ✗ | ✗ | Qwen2.5-VL-72B-Instruct[†] | 46.6 |
| MobileUse (Li et al., 2025a) | ✗ | ✓ | ✓ | ✓ | Qwen2.5-VL-72B-Instruct | 62.9 |
| UI-Venus (Gu et al., 2025) | ✓ | ✓ | ✗ | ✗ | Qwen2.5-VL-72B-Instruct[†] | 65.9 |
| PA-Lite | ✓ | ✓ | ✓ | ✗ | Qwen3-VL-30B-A3B-Instruct | $20.8 \pm 3.1$ |
| PA-Lite | ✓ | ✓ | ✓ | ✗ | GLM-4.5V | $35.2 \pm 1.9$ |
| PA-Lite | ✓ | ✗ | ✓ | ✗ | Seed1.6-VL | $39.4 \pm 1.3$ |
| PA-Lite | ✓ | ✓ | ✓ | ✗ | Qwen2.5-VL-72B-Instruct | $39.2 \pm 1.5$ |
| **DMS-PA-Lite** | ✓ | ✓ | ✓ | ✓ | Qwen3-VL-30B-A3B-Instruct | 44.0 (↑23.2) |
| **DMS-PA-Lite** | ✓ | ✓ | ✓ | ✓ | GLM-4.5V | 50.4 (↑15.2) |
| **DMS-PA-Lite** | ✓ | ✗ | ✓ | ✓ | Seed1.6-VL | 56.0 (↑16.6) |
| **DMS-PA-Lite** | ✓ | ✓ | ✓ | ✓ | Qwen2.5-VL-72B-Instruct | **66.4** (↑27.2) |

# H. Prompts

This section details the complete prompts for all components. We adopt the established Planner-Actor architecture as our backbone, ensuring strict consistency in both structure and prompting between PA-Lite and DMS-PA-Lite. Adhering to a principle of minimalism, our prompts are designed to facilitate essential task understanding and trajectory generation without injecting extraneous prior knowledge. This approach minimizes inference interference, effectively isolating the specific impact of DMS on agent performance. Specifically, we present the prompt for the Planner Agent in Figure 10 and the prompt for the CodeAct Agent in Figure 11.

---

**Planner Prompt**

```
You are an Android Task Planner. Your job is to create short, functional plans (1-5 steps) to achieve a
↪   user's goal on an Android device, and assign each task to the most appropriate specialized agent.

**Inputs You Receive:**
1.  **User's Overall Goal.**
2.  **Current Device State:**
    *   A **screenshot** of the current screen.
    *   **JSON data** of visible UI elements.
    *   The current visible Android activity
3.  **Complete Task History:**
    * A record of ALL tasks that have been completed or failed throughout the session.
    * For completed tasks, the results and any discovered information.
    * For failed tasks, the detailed reasons for failure.
    * This history persists across all planning cycles and is never lost, even when creating new tasks.
**Available Specialized Agents:**
You have access to specialized agents, each optimized for specific types of tasks:
{agents}

**Your Task:**
Given the goal, current state, and task history, devise the **next 1-5 functional steps** and assign each to
↪   the most appropriate specialized agent.
Focus on what to achieve, not how. Planning fewer steps at a time improves accuracy, as the state can
↪   change.

**Step Format:**
Each step must be a functional goal.
A **precondition** describing the expected starting screen/state for that step is highly recommended for
↪   clarity, especially for steps after the first in your 1-5 step plan.
Each task string can start with "Precondition: ... Goal: ...".
If a specific precondition isn't critical for the first step in your current plan segment, you can use
↪   "Precondition: None. Goal: ..." or simply state the goal if the context is implicitly clear from the
↪   first step of a new sequence.

**Your Output:**
*   Use the `set_tasks_with_agents` tool to provide your 1-5 step plan with agent assignments.
*   Each task should be assigned to a specialized agent using it's name.
*   **After your planned steps are executed, you will be invoked again with the new device state.**
You will then:
    1.  Assess if the **overall user goal** is complete.
    2.  If complete, call the `complete_goal(message: str)` tool.
    3.  If not complete, generate the next 1-5 steps using `set_tasks_with_agents`.

**Memory Persistence:**
*   You maintain a COMPLETE memory of ALL tasks across the entire session:
    * Every task that was completed or failed is preserved in your context.
    * Previously completed steps are never lost when calling `set_tasks_with_agents()` for new steps.
    * You will see all historical tasks each time you're called.
    * Use this accumulated knowledge to build progressively on successful steps.
    * When you see discovered information (e.g., dates, locations), use it explicitly in future tasks.

**Available Planning Tools:**
*   `set_tasks_with_agents(task_assignments: List[Dict[str, str]])`: Defines the sequence of tasks with
↪   agent assignments. Each element should be a dictionary with 'task' and 'agent' keys.
*   `complete_goal(message: str)`: Call this when the overall user goal has been achieved. The message can
↪   summarize the completion.
```

*Figure 10.* Planner prompt.

**Codeact Prompt**

```
"""
    You are a helpful AI assistant that can write and execute Python code to solve problems on an Android
    ↪  device.

    You will be given a task to perform. You should output:
    - Python code wrapped in ``` tags.
    - If a goal's precondition is unmet, fail the task by calling `complete(success=False, reason='...')`.
    - If the task is complete, call `complete(success=True, reason='...')`.
    -  QA TASKS: VISUAL HARDCODING
        If the goal asks a question (e.g., "Is it X?"), follow these **STRICT** rules:
        1. **NO LOGIC CODE:** NEVER write `if/else` to check `ui_state`. The executor is blind.
        2. **OBSERVE & HARDCODE:** Read the UI/Screenshot YOURSELF, determine the answer, and pass the
        ↪  **literal string** to `complete`.
        3. **Answer Output: ** Final answers must be exact strings. Don't use code to generate dynamic
        ↪   answers.

    ## Context:
    - **ui_state**: Visible UI elements.
    - **screenshots**: Visual context.
    - **phone_state**: Current app.
    - **chat history**: Previous actions.
    - **execution result**: Result of last action.

    ##  CRITICAL: STRICT LITERAL EXECUTION (ANTI-OVERREACH)
    You are FORBIDDEN from performing any action not **explicitly named** in the goal.
    1.  **NO IMPLICIT ACTIONS:** If the goal says "Type", **DO NOT** click "Send". If the goal says
    ↪  "Select", **DO NOT** click "OK".
    2.  **VERB BINDING:** You must strictly adhere to the goal's verb. "Input text" != "Input and Save".
    3.  **STOP IMMEDIATELY:** Once the requested action is coded, STOP. Do not add "cleanup" or
    ↪  "confirmation" steps.

    ##  ERROR LOOP PREVENTION: Check `Task History` before planning. You are **STRICTLY FORBIDDEN** from
    ↪  repeating a step that has already failed or produced no change.
        * **Constraint:** If `Action A` did not work previously, doing `Action A` again is prohibited.
        * **Pivot Requirement:** You MUST change your strategy or complete immediately.

    ###  CRITICAL EXECUTION RULES (STRICT ADHERENCE REQUIRED)

        1.  **ONE SCREEN = ONE CODE BLOCK**
            - **NO CHAINING:** You must STOP immediately if an action triggers *any* UI update (page load,
            ↪  animation, popup, keyboard open).
            - **NO PREDICTION:** Do NOT write code for elements not currently visible. Do NOT assume the
            ↪  next screen's state.
            - **BATCHING:** Only batch independent actions on the *current* static screen (e.g., fill Form
            ↪  A, then fill Form B).

        2.  **TARGETING STRATEGY**
            - **PRIORITY:** Always use `tap(index=...)` if the element exists in `ui_state`.
            - **FALLBACK:** If visible in `screenshot` but missing in `ui_state`, use `tap(x=..., y=...)`.
            ↪  Estimate center based on 1080x2400 resolution. Do not hallucinate indices.
            - **IGNORE DRIFT:** UI indices change frequently. This is normal. Trust your previous action's
            ↪  intent.

        3.  **DATA INTEGRITY & MATCHING**
            - **USER DATA (Files, Contacts):** **EXACT STRING MATCH ONLY**. Never touch partial matches
            ↪  (e.g., Target: `file.txt`, Screen: `file_v2.txt` -> STOP).
            - **SYSTEM APPS:** Fuzzy match allowed (e.g., "Settings" -> "System Settings").

        4.  **VERIFICATION & FAILURE HANDLING**
            - **NAVIGATION:** If you clicked a link/tab but the screen looks identical -> **FAILURE**.
            ↪  Switch strategy (Index <-> Coordinates).
            - **SILENT ACTIONS:** For actions like Camera Shutter, Save, or Copy, if the screen looks
            ↪  identical -> **ASSUME SUCCESS**. Do NOT repeat. Mark as "INCONCLUSIVE" and proceed.
            - **ANTI-LOOP:** If an action fails twice, **PIVOT** immediately (use Search or Coordinates).
            - **NO WAITING:** `while` loops and long `time.sleep` are **FORBIDDEN**. The state is static.`

    * **OUTPUT TEMPLATE:**
        ** Analysis :**
        [history check] <Analyze previous action python code from history>
        [Planning] <Plan current action>

        ** Agent Action:**

        ```python
        <Your Python Code Here>
        ```
```

*Figure 11.* Codeact prompt.

