# OpenReview forum: "Darwinian Memory: A Training-Free Self-Regulating Memory System for GUI Agent Evolution"
_ICML.cc/2026/Conference — ICML 2026 regular_

### Official Review · Reviewer_ETv2 · 2026-03-12

**Soundness:** 3
**Presentation:** 3
**Significance:** 3
**Originality:** 3
**Overall Recommendation:** 4
**Confidence:** 4

**Summary:**

The paper studies the problem of enabling multimodal large language model (MLLM) agents to perform long-horizon GUI automation tasks across multiple applications, where limited context windows and static memory accumulation often lead to outdated or noisy experiences that degrade performance. To address this, the authors propose Darwinian Memory System (DMS), a training-free memory architecture that models agent memory as an evolving ecosystem governed by a survival-of-the-fittest principle. The framework decomposes interaction trajectories into reusable memory units, separating high-level intent from execution states to support compositional reuse across tasks. It further introduces a utility-driven natural selection mechanism that evaluates the usefulness of stored experiences, retaining beneficial strategies while pruning suboptimal or risky trajectories to prevent context pollution. The proposed system operates without additional model training or architectural modification and can be integrated with existing MLLM-based GUI agents. Experiments on multi-application GUI benchmarks demonstrate improvements in task success rate, execution stability, and latency, suggesting that evolutionary memory management can enhance the reliability and efficiency of GUI agents.

**Compliance With Llm Reviewing Policy:**

Affirmed.

**Final Justification:**

Even after the rebuttal and clarifications, some of my concerns still remain, particularly regarding aspects like memory irreversibility and robustness under changing environments.

That said, the authors have provided reasonable explanations and additional details that strengthen the work overall. I believe the paper still meets the acceptance bar, so I am keeping my original score.

**Key Questions For Authors:**

1. The Darwinian Memory System relies on a utility-driven selection mechanism to determine which memory units survive or are pruned. Could the authors provide a more precise description of how the utility score is computed, including the exact features, weighting scheme, and thresholds used for selection?

2.The proposed framework contains several components, including trajectory decomposition, utility-based evaluation, and evolutionary pruning. Could the authors provide additional ablation studies isolating the contribution of each component?

3. The paper claims that the evolutionary pruning mechanism helps control memory growth. Could the authors provide more detailed analysis on how the memory size evolves over longer horizons, such as across many tasks or extended interaction periods?

4. The experiments focus on multi-application GUI tasks. How well does the Darwinian Memory System generalize to other agent settings, such as web navigation, embodied agents, or different multimodal models?

5. Several prior works explore memory filtering, trajectory prioritization, or episodic memory retrieval for LLM agents. Could the authors clarify how the proposed approach differs from or improves upon these existing memory management techniques, both conceptually and empirically?

**Limitations:**

Yes. The authors discuss several limitations of their approach, including the dependence of the memory selection mechanism on heuristic utility scoring and the current evaluation being limited primarily to GUI automation tasks. They acknowledge that broader validation across additional environments and agent architectures would further strengthen the generality of the proposed framework. The paper also notes that the evolutionary pruning mechanism may require careful tuning of thresholds and evaluation criteria to avoid discarding potentially useful experiences.

For further improvement, the authors could expand the discussion by addressing potential risks associated with autonomous GUI agents, such as unintended actions in real-world software environments, misuse in automated interaction systems, or privacy concerns when storing interaction trajectories. A more explicit discussion of safeguards, deployment considerations, and responsible use cases would strengthen the societal impact section.

**Strengths And Weaknesses:**

Soundness.
The paper proposes a technically coherent framework for managing memory in multimodal large language model (MLLM) agents operating in long-horizon GUI environments. The core idea—treating agent memory as an evolving system where useful trajectories are retained and less effective ones are pruned—is well motivated by the practical limitations of static memory accumulation and context window constraints. The proposed Darwinian Memory System (DMS) decomposes trajectories into reusable units and applies a utility-based selection mechanism to regulate memory growth. The design is conceptually consistent and compatible with existing MLLM agents without requiring additional training. The empirical evaluation on multi-application GUI tasks demonstrates improvements in task success rate, execution stability, and latency, suggesting that the proposed memory management mechanism can improve agent performance in complex interaction scenarios.

Although the framework is conceptually coherent, some aspects of the methodology would benefit from deeper analysis and validation. The selection and scoring mechanisms for memory units are largely heuristic, and the paper provides limited theoretical justification for the specific design choices. Additionally, while the experiments show improvements on the evaluated benchmarks, the evaluation scope is somewhat limited, making it difficult to assess the robustness of the approach across different environments, tasks, or agent architectures. More extensive ablation studies and statistical analysis could strengthen the empirical claims.

Presentation.
The paper is generally clearly written and structured. The evolutionary analogy provides an intuitive framework for understanding the system, and the authors describe the motivation and architecture of the Darwinian memory mechanism in a logical sequence. The paper explains how trajectories are decomposed, evaluated, and selectively retained, which helps readers follow the proposed workflow. The experimental results are presented in a structured manner and illustrate the potential benefits of the approach. Overall, the narrative is accessible and communicates the main ideas effectively.

While the overall structure is clear, certain implementation details and algorithmic components could be described more precisely to improve reproducibility. For instance, the criteria used to evaluate memory utility and the thresholds used for pruning are not always fully specified. A more formal description of the algorithm and clearer explanations of parameter settings would help readers better understand and reproduce the proposed method. The related work discussion could also more explicitly differentiate the proposed framework from other memory-augmented agent systems.

Significance.
The work addresses an important challenge in LLM-based agents: managing memory for long-horizon tasks where naive accumulation of past trajectories can degrade performance. As GUI automation and multimodal agents become increasingly relevant, mechanisms for scalable and reliable memory management are necessary. By proposing a training-free framework that can be integrated with existing agents, the paper offers a practical approach that may be useful for real-world deployments and future research on memory-augmented agents.

Although the problem addressed is relevant, the empirical evaluation focuses primarily on GUI automation tasks. This relatively narrow application domain may limit the perceived generality of the approach. Demonstrating the framework on a broader range of agent tasks—such as web navigation, embodied interaction, or planning benchmarks—would strengthen the argument that the proposed memory system has broader impact across machine learning and autonomous agent research.

Originality.
The paper introduces a novel perspective by framing memory management as a form of evolutionary selection over past experiences. While the individual components (trajectory storage, filtering, and reuse) are related to existing memory-based approaches, the combination of trajectory decomposition, utility-driven evaluation, and evolutionary pruning provides a distinct framework for regulating memory in agent systems. The Darwinian perspective offers a creative conceptual lens that may inspire further exploration of adaptive memory mechanisms in autonomous agents.

While the Darwinian framing is conceptually appealing, some elements of the approach resemble existing strategies in memory-based learning and experience replay, such as trajectory filtering, prioritization, and pruning. As a result, the novelty lies more in the combination and framing of these ideas rather than in fundamentally new algorithmic mechanisms. Clarifying the distinctions between the proposed system and prior memory management techniques would help better highlight the unique contributions of the work.

---

> ### Author Rebuttal · Authors · 2026-03-30
>
> Thank you for your valuable time and prompt response.
>
> ---
>
> Q1:Precision of Utility Computation
>
> We appreciate the attention to the DMS memory mechanism, as it reflects ICML’s profound emphasis on innovation. Retention scores are the product of three dimensions: reuse frequency $(ln(1+n)+V_{new​})$, temporal decay (Sigmoid-based), and a reliability penalty $(1/(1+γ⋅strike))$. The frequency component balances new and high-frequency tasks, while the decay half-life $(t_{half​}=t_{base​}+α⋅ln(1+n))$ extends with reuse to prioritize verified experiences. The penalty rapidly degrades scores for trajectories failing verification. For selection, the Gradient Elbow Method identifies inflection points via the score distribution's second derivative, enabling adaptive capacity regulation instead of static thresholds. This maintains storage at ~5MB while ensuring trajectory consistency. Detailed parameters (α=15,β=0.5) are documented in Appendix B for reproducibility.
>
> ---
>
> Q2: Contribution of Each Component
>
> We thank the reviewer for the suggestion to isolate component contributions, as addressed in Table 2 and Figure 8 of our manuscript. DMS performance is driven by the synergy of trajectory decomposition, self-regulation (SR), and feedback verification. Decomposition is fundamental: Table 2 shows that replacing it with a Standard Goal-Based Key drops the success rate (SR) to 29.7%, underperforming the memory-free PA-Lite baseline (41.0%) due to negative transfer. SR and feedback ensure precision; removing SR drops SR to 39.2% as disordered memory triggers false-positive hallucinations, while feedback ablation prevents purging invalidated experiences. Efficiency gains are equally significant: Fig. 8 shows full DMS reduces execution time by 18.8% (164.8s vs. 203.4s without regulation). These components ensure high storage precision and execution robustness for long-term GUI automation.
>
> ---
> Q3: Evolution of Memory Size Over Long Horizons
>
> The reviewer’s inquiry into memory size evolution is highly professional, reflecting the performance of our work in practical engineering applications. Across 5 rounds (580 tasks), DMS utilized evolutionary pruning to maintain stability. Following initial skill acquisition, memory size enters a state of controlled oscillation rather than linear expansion. Pruning is triggered via the Gradient Elbow Method when the survival score distribution identifies low-value entries. For Qwen2.5-VL-72B, maintenance cycles reduced 113 entries to a baseline of 50 at Step 150, keeping the footprint between 5MB and 8MB. As shown in Fig. 8 (left), the DMS storage slope flattens as the system matures, ensuring millisecond-level retrieval and preventing performance degradation. By eliminating redundant trajectories, DMS maintains a high-density skill set for indefinite interaction sequences.
>
> ---
> Q4: Generalization to Other Settings
>
> The reviewer’s perspective is enlightening, as generalization dictates the functional scope of DMS within the research community. DMS is a model-agnostic, training-free framework that generalizes across various LMMs, including Qwen2.5-VL and GPT-4o. Its performance gains stem from structured reuse rather than specific model weights, consistently reducing inference burden regardless of the backend’s inherent reasoning capacity. This plug-and-play architecture ensures robustness across models of different scales and architectures.
> DMS generalizes to web and embodied AI: hierarchical decomposition (e.g., DOM trees) mitigates web noise, while pruning/verification minimize embodied trial costs. Due to space limits, we address broader generalization in Reviewer ksnV’s Q1. As noted in Appendix C, fitness-based selection serves as a universal meta-strategy for long-term decision-making, providing a scalable knowledge solution for open-domain interactions.
>
> ---
> Q5: Comparison with Prior Memory Management Techniques
>
> DMS distinguishes itself through its evolutionary capability, treating the memory bank as a dynamic, self-refining system rather than a passive repository. Unlike passive models that expand linearly causing retrieval interference, DMS employs an active selection process based on a multi-dimensional fitness score (utility, decay, reliability). Empirically, the Gradient Elbow Method adaptively prunes low-quality trajectories, resolving performance degradation. Results show DMS reduced execution latency by 18.8% (164.8s vs. 203.4s) while stabilizing storage at ~5MB. Additionally, trajectory decomposition prevents negative transfer by enabling flexible sub-goal reuse instead of failed monolithic matching. Ultimately, DMS’s bio-inspired logic provides a self-purifying solution to memory overload challenges in autonomous agents.
>
> Finally, thank you again for your valuable time and constructive comments. For a researcher, the happiest thing is to feel that the work is understood by others. Thanks a lot!

---

> > ### Author Rebuttal · Reviewer_ETv2 · 2026-04-03
> >
> > The authors have addressed my earlier questions satisfactorily, particularly by clarifying the utility computation, providing ablation results, and discussing memory scaling behavior.
> >
> > However, a few concerns remain:
> >
> > Memory irreversibility:
> > The current framework appears to permanently discard pruned memory units. It is unclear whether the system can recover or reintroduce previously removed trajectories if similar tasks reappear later. In non-stationary environments, trajectories that were initially low-utility may become useful over time. Could the authors clarify how such scenarios are handled?
> >
> > Robustness to GUI changes:
> > The method relies on heuristic scoring and trajectory decomposition tied to past interaction patterns. In dynamic GUI environments where layouts and workflows frequently change, there is a risk that the memory becomes over-specialized to a particular interface. How does the system maintain robustness under such distribution shifts?

---

> > > ### Author Response · Authors · 2026-04-05
> > >
> > > We appreciate the reviewer’s constructive follow-up and the positive recognition of our work. Below, we provide our detailed responses to the newly raised inquiries:
> > >
> > > ---
> > > Q1: Memory irreversibility
> > >
> > > The reviewer’s inquiry regarding memory irreversibility is profoundly insightful, particularly concerning adaptation in non-stationary environments. DMS physically purges low-utility units from the long-tail distribution to maintain a high-density active bank, but this process is an active state update rather than a permanent loss of system capability. In dynamic GUI environments, pruning is essential to minimize retrieval interference from obsolete or sub-optimal trajectories that could otherwise lead to "memory poisoning."
> > >
> > > If a previously pruned task reappears or a scenario becomes relevant again due to environmental shifts, DMS naturally triggers a new "exploration-feedback-archiving" cycle. Rather than simply restoring a potentially outdated memory, the agent re-evolves a trajectory precisely adapted to the current environmental state. As noted by reviewer, GUI environments are inherently non-stationary; direct reuse of historical memory entails significant risks and prohibitive verification costs. Furthermore, maintaining a vast volume of long-unused memory incurs unnecessary storage and maintenance overhead.
> > >
> > > Ultimately, DMS prioritizes retaining high-value, verified knowledge with minimal storage costs. Our pruning strategy accelerates the removal of low-value entries, making room for high-quality trajectories that better align with the current environment. This cycle of early detection, rapid correction, and autonomous re-learning ensures DMS remains agile and reliable, even amidst significant environmental transitions.
> > >
> > > ---
> > >
> > > Q2: Robustness to GUI changes
> > >
> > > Regarding the robustness to GUI changes, DMS is architected as a dynamic, self-correcting memory system rather than a static repository of past interaction patterns. To mitigate the risk of over-specialization, our primary defense is the ϵ-mutation mechanism, which maintains a strategic level of exploration even when high-utility memories are available. This ensures the agent remains sensitive to environmental shifts and is capable of discovering new optimal paths as GUI layouts or workflows evolve. When significant distribution shifts occur, trajectories that no longer align with the updated interface will fail the feedback-verification stage. These failing units are then rapidly penalized and purged via the K-Verification strategy (detailed in Appendix C), effectively preventing the system from re-applying obsolete patterns to new environments.
> > >
> > > Furthermore, trajectory decomposition in DMS focuses on functional sub-goals, providing a level of abstraction that enhances robustness against structural changes. Rather than being anchored to rigid, coordinate-based sequences, the system preserves the underlying task logic.When encountering GUI shifts, the agent utilizes the LLM’s reasoning to adapt its execution of sub-goals, while still benefiting from the structured logic stored in memory. This balance between high-level logical reuse and low-level adaptive execution allows DMS to adapt across various GUI versions. Consequently, the Darwinian process of continuous selection and mutation ensures the memory bank remains in a state of "fluid stability," constantly shedding specialized biases in favor of trajectories with the highest current utility.
> > >
> > > ---
> > >
> > > We sincerely appreciate the reviewer’s time and the depth of engagement with our manuscript. We hope these clarifications effectively address your concerns.

---

### Official Review · Reviewer_GL9U · 2026-03-12

**Soundness:** 2
**Presentation:** 2
**Significance:** 3
**Originality:** 3
**Overall Recommendation:** 4
**Confidence:** 4

**Summary:**

The paper proposes the Darwinian Memory System (DMS), a training-free, self-evolving memory framework designed for GUI agents. To address the rigidity and context pollution of existing memory paradigms, DMS deconstructs task trajectories into reusable sub-plans and employs a "survival of the fittest" selection mechanism. This system incorporates utility-driven pruning, a Bayesian risk assessment for plan inhibition, and a mutation mechanism to encourage exploration. Evaluated on the AndroidWorld benchmark, the authors report significant improvements in success rate and execution stability across several general-purpose MLLMs. Overall, this study's primary area is the development of self-regulating memory architectures for multimodal GUI agents.

**Compliance With Llm Reviewing Policy:**

Affirmed.

**Final Justification:**

I have read the authors’ rebuttal and the subsequent discussion. The rebuttal adequately addressed my main concerns. Accordingly, I have raised my score by 1 point.

**Key Questions For Authors:**

1. **Hyperparameter Transparency:** What is the value of the mutation probability $\epsilon$ used in the experiments? Please provide a sensitivity analysis of $\epsilon$ to demonstrate its impact on the exploration-exploitation balance.
2. **Retrieval Ablation:** Why was the multiplicative similarity score chosen over a simpler additive score or a weighted sum? Does the multiplicative form lead to higher precision in memory retrieval?
3. **Evaluation Breadth:** How does DMS perform on non-Android GUI benchmarks like WebArena or OSWorld? Can the authors provide at least a preliminary cross-domain evaluation?
4. **Efficiency Overhead:** What is the average wall-clock overhead introduced by the Verifier Agent and the memory regulation (pruning) step? Does this overhead negate the latency gains from trajectory reuse?
5. **Trajectory Filtering:** Can you provide an ablation study for the $|\tau| > 1$ filter? Specifically, how many "Easy" tasks in AndroidWorld were negatively impacted by discarding single-step solutions?

If the above questions are resolved, my score will increase.

**Limitations:**

Yes

**Strengths And Weaknesses:**

**Strengths:**
1. **Practicality:** The method is training-free and model-agnostic, making it easily deployable on top of existing MLLMs without expensive fine-tuning.
2. **Granularity:** By decomposing monolithic trajectories into atomic sub-plan units, the system offers better flexibility in dynamic GUI environments compared to static trajectory replay.
3. **Efficiency:** The use of retrieved trajectories as "macros" successfully reduces inference latency and improves deterministic execution in repeated sub-tasks.

**Weaknesses:**

**1. Soundness of Methodological Choices and Hyperparameters**
Several critical design choices lack empirical or theoretical justification.
*   **Retrieval Formula:** The multiplicative similarity score $\text{sim}(\phi(\tilde{p}_\text{pre}), \phi(p_\text{pre})) \cdot \text{sim}(\phi(\tilde{p}_\text{goal}), \phi(p_\text{goal}))$ is used without justification. This product-based approach is sensitive to low scores in either term; the authors do not ablate this against simpler additive or weighted alternatives.
*   **Trajectory Filtering:** The exclusion of $|\tau|=1$ (single-step) trajectories is claimed to ensure "non-trivial behavior," yet no ablation is provided. This may discard optimal one-step solutions for simple GUI tasks, potentially biasing the results toward complex tasks.
*   **Missing Hyperparameter:** The mutation probability $\epsilon$, which governs the core exploration-exploitation trade-off, is neither reported in the hyperparameter table nor ablated. This omission hinders reproducibility.

**2. Presentation and Scientific Metaphor**
*   **Misleading Metrics:** The reported "average" improvements in success rate (18.0%) and stability (33.9%) are misleading. As shown in Figure 5, performance improves monotonically over rounds. Averaging results across the learning curve conflates the "learning process" with the "steady-state performance," making it difficult to discern the true evolved capability of the system.
*   **Inaccurate Metaphor:** The biological metaphor is scientifically loose. Darwinian evolution requires population dynamics and genetic recombination, which are absent here. The term "survival of the fittest" is applied to sub-task trajectories within a single-agent session, which is more akin to reinforcement learning with experience replay than evolutionary biology.
*   **Vague Experimental Setup:** The "multi-app" evaluation in Figure 1 lacks formal description, including task counts, specific protocols, and the baseline definitions used for comparison.

**3. Significance and Evaluation Scope**
*   **Single-Domain Evaluation:** The evaluation is restricted to AndroidWorld. Given the title's claim regarding "GUI Agent Evolution," the lack of testing on other critical domains such as Web (e.g., WebArena[3]) or Desktop (e.g., OSWorld[4]) limits the evidence for the system's generalizability.
*   **Incomplete Latency Analysis:** While the authors claim reduced task latency, the analysis excludes the computational overhead of the Verifier Agent and the embedding updates required for memory maintenance. A wall-clock time comparison inclusive of all DMS components is missing.

**4. Originality and Positioning**
The paper identifies two bottlenecks—rigidity and context pollution—that have been extensively explored in recent literature (e.g., EchoTrail-GUI[1], MemEvolve[2]). The authors fail to sufficiently differentiate DMS from these directly competing works in the introduction. Overall, the authors assess a central concept of using evolutionary-inspired pruning to manage GUI memory, but the novelty relative to existing dynamic replay systems needs clearer articulation.

[1] Li et al., EchoTrail-GUI: Building Actionable Memory for GUI Agents via Critic-Guided Self-Exploration.

[2] Zhang et al., MemEvolve: Meta-Evolution of Agent Memory Systems.

[3] Zhou et al., WebArena: A Realistic Web Environment for Next-Generation Agents.

[4] Xie et al., OSWorld: Benchmarking Multimodal Agents for Open-Ended Operating System Orchestration.

---

> ### Author Rebuttal · Authors · 2026-03-30
>
> We sincerely thank the reviewer for the time spent carefully reading our manuscript and for providing such valuable and forward-looking feedback.
>
> ---
>
> Q1: Hyperparameter Transparency
>
> We sincerely thank the reviewer for pointing out this omission. The mutation probability ϵ=0.1 is set as a conservative baseline to prioritize stability and efficiency in large-scale GUI automation, ensuring consistent performance while facilitating evolutionary dynamics. Due to the rebuttal timeframe, executing 2,900 live tasks (116×5×5) was infeasible; we analyzed a stratified random sample of 29 tasks (25% of the benchmark, preserving original difficulty). Results using Qwen2.5-VL-72B are:
>
> Ablation on ϵ (SR/RR):
>
> ϵ=0.1: 0.52/0.09, 0.73/0.22, 0.66/0.28, 0.66/0.22, 0.68/0.34;
>
> ϵ=0.2: 0.57/0.08, 0.57/0.17, 0.66/0.25, 0.66/0.23, 0.66/0.21;
>
> ϵ=0.3: 0.57/0.04, 0.61/0.13, 0.61/0.12, 0.64/0.15, 0.66/0.16;
>
> ϵ=0.4: 0.52/0.01, 0.57/0.13, 0.57/0.05, 0.61/0.04, 0.61/0.16;
>
>  ϵ=0.6: 0.52/0.01, 0.52/0.03, 0.36/0.06, 0.41/0.02, 0.34/0.02.
>
> Data reveals a critical exploration-stability trade-off. While higher rates (ϵ≥0.2) slightly aid early exploration (e.g., R1), they hinder long-term consolidation. Conversely, ϵ=0.1 offers the optimal balance. It retains sufficient mutation to escape local optima (precluding ϵ→0) while achieving the highest steady-state performance by Round 5 (SR 0.68/RR 0.34). Rates ≥0.3 disrupt memory and force zero-shot reasoning. We recommend ϵ∈[0.1,0.2] for stable deployment and have added these details to Appendix B.
>
> ---
>
> Q2: Retrieval Ablation Logic
>
> Multiplicative scoring was chosen over additive sums to eliminate the compensation effect. In GUI automation, mismatched preconditions render trajectories non-executable regardless of goal similarity. While additive models allow high goal matches to mask precondition failures, the multiplicative form imposes a non-linear penalty—effectively a Logical AND gate.
>
> This magnifies contrast: for 0.9 goal similarity, 0.8 vs 0.4 precondition scores yield a 2.0x margin (0.72 vs 0.36) via multiplication, compared to 1.3x (1.7 vs 1.3) via addition. This steeper gradient ensures precise selection. Table 2 in our manuscript confirms that removing preconditions (diluting constraints) drops success rates to 29.7%, far below the 41.0% PA-Lite baseline. This design effectively enhances retrieval by tightly coupling algorithmic logic with practical GUI task characteristics, thereby suppressing false positives and hallucinations.
>
> ---
>
> Q3: Cross-Platform Generalization
>
> We highly value this insightful suggestion. Due to space constraints and identical concerns, please refer to our detailed response in Reviewer ksnV’s Q1, where we provide a comprehensive discussion.
>
> ---
>
> Q4: Wall-Clock Overhead Analysis
>
> We highly value the reviewer’s astute focus on operational efficiency, as it pertains directly to the critical engineering strategies for the actual system implementation. DMS uses an asynchronous, non-blocking architecture where regulation (scoring and pruning) is offloaded to a separate daemon thread. This ensures no perceptible latency or bottleneck to the main execution loop.
>
> Regarding the Verifier Agent, quantitative analysis on Qwen2.5-VL-72B shows verification overhead for 116 tasks is just 37.12s—a mere 0.23% of total duration and negligible compared to gains in Fig. 1. Furthermore, Fig. 8 shows average per-task latency drops from 203.4s to 164.8s with DMS. This 18.8% reduction confirms the Darwinian Memory system’s positive net efficiency gain.
>
> ---
>
> Q5: Ablation of the ∣τ∣>1 Filter
>
> The ∣τ∣>1 filter is a storage optimization: it determines if a trajectory is "worthy" of archiving without impacting the agent's ability to solve atomic tasks via MLLM reasoning. In the ablation study across 3,800 steps, ∣τ∣=1 trajectories (e.g., "Tap OK") showed a 1.75% Reuse Coverage; 98.25% were eventually pruned.
>
> | ∣τ∣ | Count | Reuse Coverage | Max Reuse | Min Reuse |
> | :--- | :--- | :--- | :--- | :--- |
> | 1 | 114 | 1.75% | 2 | 0 |
> | 2 | 1724 | 13.28% | 76 | 0 |
> | 3 | 349 | 10.32% | 46 | 0 |
> | 4 | 136 | 16.91% | 14 | 0 |
> | ... | ... | ... | ... | ... |
> | 10 | 14 | 7.14% | 12 | 0 |
>
> Since the cost of retrieval/verification for atomic steps yields near-zero marginal gain compared to MLLM inference, filtering these ensures the memory bank remains a collection of high-density structural skills, directly enhancing long-term stability and efficiency. This approach champions a lean knowledge paradigm, aligning with the industrial shift toward high-efficiency mobile automation.
>
> ---
>
> We thank the reviewer for meticulous feedback reflecting ICML’s commitment to innovation. This response results from days of continuous, intensive work by all authors. We hope this addresses your concerns and would appreciate a re-evaluation; your feedback has been invaluable in refining our manuscript.

---

> > ### Author Rebuttal · Reviewer_GL9U · 2026-04-01
> >
> > I would like to sincerely thank the authors for their highly diligent, comprehensive, and convincing rebuttal. The amount of work done during this short rebuttal period—especially running the ablation studies on a 72B model—is highly appreciated and commendable.
> >
> > I am very happy to raise my score to support the acceptance of this work.

---

> > > ### Author Response · Authors · 2026-04-07
> > >
> > > We sincerely thank the reviewer for the encouraging feedback. Your thoughtful guidance has been instrumental in refining this work, and we are deeply grateful for your support.

---

### Official Review · Reviewer_ksnV · 2026-03-13

**Soundness:** 2
**Presentation:** 3
**Significance:** 2
**Originality:** 3
**Overall Recommendation:** 4
**Confidence:** 3

**Summary:**

This paper proposes the Darwinian Memory System (DMS), a training-free memory architecture for GUI agents that draws inspiration from evolutionary biology. DMS addresses two core failure modes in existing memory systems, rigidity of monolithic trajectory storage and accumulation of outdated/toxic memories, by decomposing workflows into reusable sub-task units and applying a survival-value-based pruning mechanism alongside Bayesian risk assessment. Experiments on AndroidWorld show consistent gains across four MLLM backbones, with average improvements of 18% in success rate and 33.9% in execution stability.

**Compliance With Llm Reviewing Policy:**

Affirmed.

**Key Questions For Authors:**

See weakness

**Limitations:**

Yes

**Strengths And Weaknesses:**

**Strengths:**
1. DMS plugs into general-purpose MLLMs without architectural changes or domain-specific pre-training, making it broadly applicable and practically appealing.
2. Strong empirical results. Achieving 66.4% on AndroidWorld with Qwen2.5-VL-72B surpasses both proprietary models and fine-tuned specialists. The multi-round evaluation protocol is more realistic than single-pass benchmarks.


**Weakness:**
1. All experiments are conducted exclusively on AndroidWorld. Evaluation on at least one additional benchmark (e.g., OSWorld, WebArena, or a desktop GUI benchmark) is necessary to substantiate the generalization claims made in the abstract and conclusion.
2. The case study reveals that 100% memory reuse led to task failure. While DMS eventually recovers, the paper does not quantify how often this occurs across all tasks, which could mask a systematic failure mode.
3. How many tasks/rounds does DMS typically require to exceed baseline performance from a cold start? Is there a principled way to estimate this for a new deployment?

---

> ### Author Rebuttal · Authors · 2026-03-30
>
> We sincerely appreciate the reviewers’ insightful comments and constructive suggestions, which have greatly helped us improve the quality of our manuscript.
>
> ---
>
> Q1: Cross-Platform Generalization
>
> We highly value the suggestion on cross-platform generalization. As Mobile and Web/OS GUI differ significantly in observation, state, and action granularity, we, consistent with many non-foundation model studies in this field [1,2,3], chose to focus on a single domain. DMS was primarily designed for Mobile GUI’s long-horizon and memory challenges, which we will clarify in the revision. However, finding the reviewer’s perspective highly enlightening, we refactored 2.7k+ lines of core code to adapt DMS to OSWorld. Limited by time, we evaluated 38 core tasks (sampled from 361) using Qwen2.5-VL-72B and Qwen3-VL-Flash over five iterations.
>
> | Model | Baseline | R1 | R2 | R3 | R4 | R5 |
> | :--- | :---: | :---: | :---: | :---: | :---: | :---: |
> | Qwen2.5-VL-72B-Instruct | 7.9% | 15.8% | 18.4% | 15.8% | 21.1% | 23.7% |
> | Qwen3-VL-Flash | 21.1% | 26.3% | 31.6% | 28.9% | 36.8% | 36.8% |
>
> Qwen2.5-VL-72B success rose from 7.9% to 23.7%, showing strong cold-start efficiency. Qwen3-VL-Flash improved from 21.1% to 36.8% by R5, solving VLC and VS Code tasks fully. MRR reached 11.8% with 100% reuse success, verifying DMS’s self-purification against hallucinations. Intermediate performance fluctuations prove the ϵ-mutation mechanism escapes local optima for convergence. Results confirm DMS is a platform-agnostic, self-evolving architecture scalable across GUI domains.
>
> [1] Li et al., Mobileuse: A gui agent with hierarchical reflection for autonomous mobile operation
>
> [2]Xie et al.,GUI-explorer: Autonomous Exploration and Mining of Transition-aware Knowledge for GUI Agent
>
> [3] Gu et al., UI-Venus Technical Report: Building High-performance UI Agents with RFT
>
> ---
> Q2: Failure Quantification during Reuse
>
> We appreciate the reviewer’s profoundly insightful observation regarding the case study. Distinguishing between transient evolutionary states and systematic failures is vital. In DMS, occasional failures during reuse are not flaws but essential triggers for the "failure-feedback-adjustment" cycle. In dynamic GUI environments, identifying these sub-optimal points allows DMS to refine and purify the memory bank against environmental shifts.
> Analyzing 580 executions (116×5), we found "reuse-then-failure" occurred in only 4 instances (0.69%). These rare edge cases were concentrated in early stages, where newly archived memories are still stabilizing. Darwinian selection effectively prunes these weak links as the process matures.
> Robustness is further validated by a Success Rate Retention (SRR) of 86.3% on Qwen2.5-VL-72B. Once a memory stabilizes, it remains overwhelmingly reliable. Our proactive "Strike-out" policy ensures that any verification failure increases an entry's strike count until it is physically purged. This prevents unreliable paths from becoming ingrained, forcing the agent to discover more robust trajectories. This cycle of early detection and rapid correction is key to DMS‘s sustained efficiency gains.
>
> ---
>
> Q3: Cold-start Efficiency and Predictability
>
> This inquiry reflects the reviewer’s profound engineering insight into the practical challenges of cold-start efficiency and predictability. Efficiency is intrinsically linked to task complexity and base model reasoning. DMS typically outperforms baselines within 2 to 3 rounds. As a training-free architecture, any success in Round 1 becomes reusable memory immediately. By Round 2, the agent bypasses heavy MLLM reasoning for similar goals, directly boosting success rates and efficiency. Figure 5a shows a clear performance leap by Round 3, proving high sample utilization for recurring GUI tasks.
> For convergence estimation in new deployments, we provide a mathematical framework in Appendix C. Improvement velocity depends on Task Recurrence Density and the initial success rate. Expected gains in Round 2 can be modeled as:
>
> $G=SR_{init}​×R_{task​}+Δv$
>
> where SR_init​ is the initial success rate, Rtask​ is the recurrence rate, and Δv is the reasoning variance correction. As interaction cycles increase, memory quality follows the stochastic process derived in Appendix C, eventually reaching a steady state. Developers can evaluate task overlap via minimal sampling and use our theoretical model to estimate the cycles required for stability. DMS is well-equipped for requirements ranging from simple single-app automation to complex, cross-application tasks, offering a robust and innovative paradigm to catalyze the community’s collective progress in this field.
>
> The above constitutes our complete response, which we hope resolves all remaining uncertainties. Your expert feedback has been invaluable, and we deeply appreciate the time and care you have devoted to reviewing our manuscript.

---

> > ### Author Rebuttal · Reviewer_ksnV · 2026-04-04
> >
> > I thank the author for the rebuttal and I would like to maintain my positive score

---

> > > ### Author Response · Authors · 2026-04-07
> > >
> > > We sincerely thank the reviewer for the continued support and for maintaining the positive assessment of our work. We are pleased that our rebuttal successfully addressed the raised points, and we appreciate your ongoing engagement in refining this manuscript.

---

### Decision · Program_Chairs · 2026-04-30

**Decision:**

Accept (regular)

**Comment:**

This submission proposes Darwinian Memory System (DMS). It is a training-free memory system for GUI agents. The main idea is to store reusable memory units and keep useful ones while removing weak ones. The method is easy to add to existing MLLM agents and does not need extra training.

There is consensus among reviewers with three weak acceptance. The Reviewers agreed that the problem is important and that the method gives clear gains on GUI tasks. The rebuttal added useful ablations, cross-domain results, and more analysis of memory behavior.

Major strengths
- 1.	The proposed method is practical and easy to use. Three reviewers agreed that DMS is training-free, model-agnostic, and easy to plug into existing GUI agents (ksnV, GL9U, ETv2).
- 2.	The  empirical results are relatively strong. The experiments showsbetter success rate, stability, and latency on AndroidWorld across several llm backbones (ksnV, GL9U, ETv2).
- 3.	During the rebuttal, the authors added results on OSWorld, ablations for key design choices, and analysis of memory growth. This addressed major reviewer concerns (ksnV, GL9U).

Main weaknesses
	- 1.	Several reviewers mentioned that its evaluation scope was limited in the main paper. Reviewers noted that the main evaluation was mostly on AndroidWorld in the original submission (ksnV, GL9U, ETv2).
	- 2. Reviewers raises quesitons on the design choices and robustness on hyperparameters, retrieval scoring, pruning, and filtering choices (GL9U, ETv2).

Overall, the recommendation is weak accept as three weak acceptance were given by the reviewers (the rebuttal strengthened the paper). The work addresses an important problem for long-horizon GUI agents and gives a practical solution that does not require training. The results are relatively strong, and the added analyses helped clarify the method. There are still some limits in evaluation scope and some open robustness questions, but overall the strengths are stronger than the weaknesses.